# Beyond Message Passing: Neural Graph Pattern Machine

**Zehong Wang** [1]  **Zheyuan Zhang** [1]  **Tianyi Ma** [1]  **Nitesh V Chawla** [1]  **Chuxu Zhang** [2]  **Yanfang Ye** [1] [*]

## Abstract

Graph learning tasks often hinge on identifying key substructure patterns—such as triadic closures in social networks or benzene rings in molecular graphs—that underpin downstream performance. However, most existing graph neural networks (GNNs) rely on message passing, which aggregates local neighborhood information iteratively and struggles to explicitly capture such fundamental motifs, like triangles, $k$-cliques, and rings. This limitation hinders both expressiveness and long-range dependency modeling. In this paper, we introduce the Neural **G**raph **P**attern **M**achine (**GPM**), a novel framework that bypasses message passing by learning directly from graph substructures. GPM efficiently extracts, encodes, and prioritizes task-relevant graph patterns, offering greater expressivity and improved ability to capture long-range dependencies. Empirical evaluations across four standard tasks—node classification, link prediction, graph classification, and graph regression—demonstrate that GPM outperforms state-of-the-art baselines. Further analysis reveals that GPM exhibits strong out-of-distribution generalization, desirable scalability, and enhanced interpretability. Code and datasets are available at: https://github.com/Zehong-Wang/GPM.

## 1. Introduction

Graphs serve as a fundamental abstraction for modeling complex systems across domains such as social networks, drug discovery, and recommender systems (Zhang et al., 2024c;d; Ma et al., 2025a; Zhang et al., 2024b; Ma et al., 2023; Fu et al., 2023). Many real-world problems can be formulated as classification or regression tasks on graphs. For example, molecular property prediction can be cast as a graph classification task, where atoms and bonds are represented as nodes and edges, respectively (Hu et al., 2020). In e-commerce, predicting user preferences naturally translates into link prediction on user-item interaction graphs (Ying et al., 2018). A key insight of these tasks is that the certain substructures encode meaningful inductive biases (Xu et al., 2019; Zhao et al., 2022), emerging as predictive patterns. For instance, triadic closure, where three nodes form a closed triangle, is ubiquitous in social, biological, and communication networks (Granovetter, 1973). It serves as an indicator of stable node relationships and is instrumental in tasks like node classification (Jin et al., 2020) and link prediction (Huang et al., 2015). In molecular graphs, the benzene ring—a six-carbon cyclic structure—is a canonical example of a stable substructure with implications for chemical reactivity (Rong et al., 2020). We refer to such recurring substructures, including triangles, rings, and other motifs, as *substructure patterns* or *graph patterns*. These patterns form the building blocks of graph semantics and are central to understanding and improving performance in graph-based learning tasks.

Despite the critical role of substructure patterns in graph learning, most graph neural networks (GNNs) (Kipf & Welling, 2017; Hamilton et al., 2017; Veličković et al., 2018) operate under the *message passing* paradigm (Gilmer et al., 2017), which iteratively aggregates local neighborhood information rather than directly modeling graph substructures. While message passing GNNs have demonstrated strong empirical performance on tasks such as node classification, link prediction, and graph classification, numerous studies (Xu et al., 2019; Verma & Zhang, 2019; Garg et al., 2020; Chen et al., 2020; Tang & Liu, 2023; Zhang et al., 2024a) highlight their inherent limitations in capturing basic patterns like triangles, stars, and $k$-cliques, owing to their equivalence to the 1-dimensional Weisfeiler-Leman (1-WL) test (Xu et al., 2019). To address these limitations, various enhancements have been proposed, including positional encodings (Murphy et al., 2019; Loukas, 2020), graph transformers (Kreuzer et al., 2021; Rampasek et al., 2022), and higher-order GNNs (Morris et al., 2019; Zhao et al., 2022; Qian et al., 2024; Ma et al., 2025b), which extend expressiveness beyond 1-WL. However, these methods remain fundamentally tied to message passing, and often suffer from challenges such as limited interpretability, high

[1]University of Notre Dame [2]University of Connecticut. Correspondence to: Zehong Wang <zwang43@nd.edu>, Yanfang Ye <yye7@nd.edu>.

*Proceedings of the 42nd International Conference on Machine Learning*, Vancouver, Canada. PMLR 267, 2025. Copyright 2025 by the author(s).

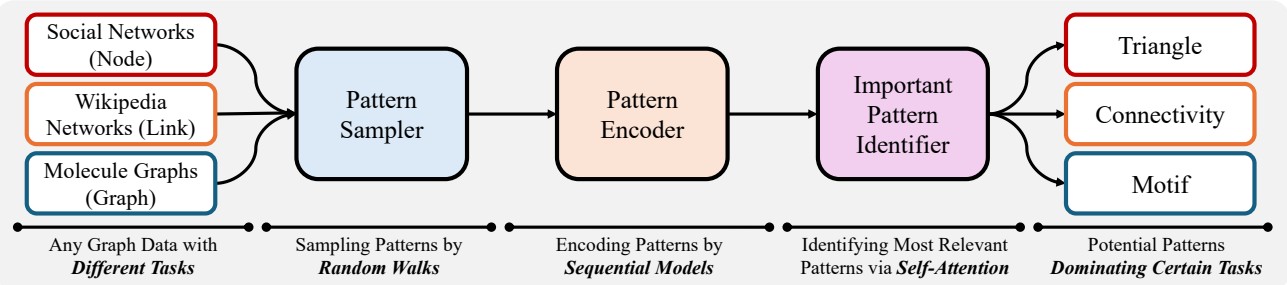

*Figure 1.* **The workflow of Neural Graph Pattern Machine (GPM).** Given a graph dataset, GPM utilizes a random walk tokenizer to extract a set of patterns representing the learning instances (nodes, edges, or graphs). These patterns are first encoded by a sequential model and then processed by a transformer encoder, which identifies the dominant patterns relevant to downstream tasks.

computational cost, and biased inductive assumptions.

To go beyond the message passing, recent research has begun exploring the use of graph patterns as discrete tokens to represent nodes, links, or entire graphs. While promising, this line of work is still in its early stages and faces three key challenges. (1) *Universal Graph Tokenizer*: Extracting graph patterns requires a tokenizer that can adapt to node-, link-, and graph-level tasks. Unlike sentences and images, which inherently have sequential structures, tokenizing graph instances is challenging due to their non-Euclidean nature. Existing tokenizers tend to be task-specific, such as the Hop2token tokenizer (Chen et al., 2023) for node tasks and the METIS tokenizer (He et al., 2023) for graph tasks. (2) *Effective Pattern Encoder*: The model must efficiently and comprehensively encode the extracted graph patterns. Current approaches often use message passing as a pattern encoder (Chen et al., 2022; He et al., 2023; Behrouz & Hashemi, 2024) or auxiliary module (Behrouz & Hashemi, 2024) to provide graph inductive biases. However, the limited expressiveness of message passing in identifying basic substructures leads to information loss in encoding graph patterns. (3) *Important Pattern Identifier*: As the sampled patterns may be duplicated or noisy, it is crucial to identify the most relevant patterns for downstream domains. For example, in social networks that favor localized patterns, the model should emphasize patterns that preserve local information. However, existing methods are generally evaluated on benchmarks with long-range dependencies (Dwivedi et al., 2023; 2022; Platonov et al., 2023); their effectiveness on graphs favoring localized or mixed dependencies remains unclear.

To address these challenges, we introduce the Neural Graph Pattern Machine (GPM), where the workflow is illustrated in Figure 1. We design a random walk-based tokenizer to sample graph patterns, which is computationally efficient (Grover & Leskovec, 2016) and can be naturally adapted to various tasks. The key insight is that *the combination of semantic path and anonymous path of a random walk*

*matches a particular graph pattern* (proved in Section 3.1). By leveraging this insight, we model graph patterns by separately encoding the semantic paths and anonymous paths of the corresponding walks, thereby comprehensively capturing the preserved graph inductive biases. The encoded graph patterns are then fed into a transformer layer (Vaswani et al., 2017) that identifies the important patterns dominating downstream tasks. To provide a deeper understanding on the superiority of GPM over message passing, we demonstrate that GPM can distinguish non-isomorphic graphs that GNNs cannot identify and can model long-range dependencies that GNNs fail to capture. Furthermore, we conduct extensive experiments on node-, link-, and graph-level tasks to demonstrate that GPM is applicable to various graph tasks and outperforms state-of-the-art baselines. Our experimental results also show that GPM is robust to out-of-distribution issues, and can be readily scaled to large graphs, large model sizes, and distributed training. In addition, the mechanism for identifying dominant substructures enables GPM to have desirable model interpretability.

## 2. Related Works

**Expressive Bottleneck of Message Passing.** Traditional message passing GNNs (Kipf & Welling, 2017; Hamilton et al., 2017; Veličković et al., 2018) suffer from well-known issues, particularly on limited expressiveness. Their expressiveness is fundamentally bounded by the 1-WL test (Xu et al., 2019; Corso et al., 2020), which restricts the ability to distinguish basic structures such as stars, cycles, and cliques (Chen et al., 2020; Garg et al., 2020; Zhang et al., 2024a; Qian et al., 2025). To address this limitation, three main research directions have emerged. First, expressive GNN variants enhance the message passing framework by incorporating higher-order structure encodings (Maron et al., 2019a; Bouritsas et al., 2022) and node positional embeddings (Murphy et al., 2019; Loukas, 2020), but often suffer from scalability challenges (Azizian & marc lelarge, 2021). Second, random walk-based approaches (Zhang

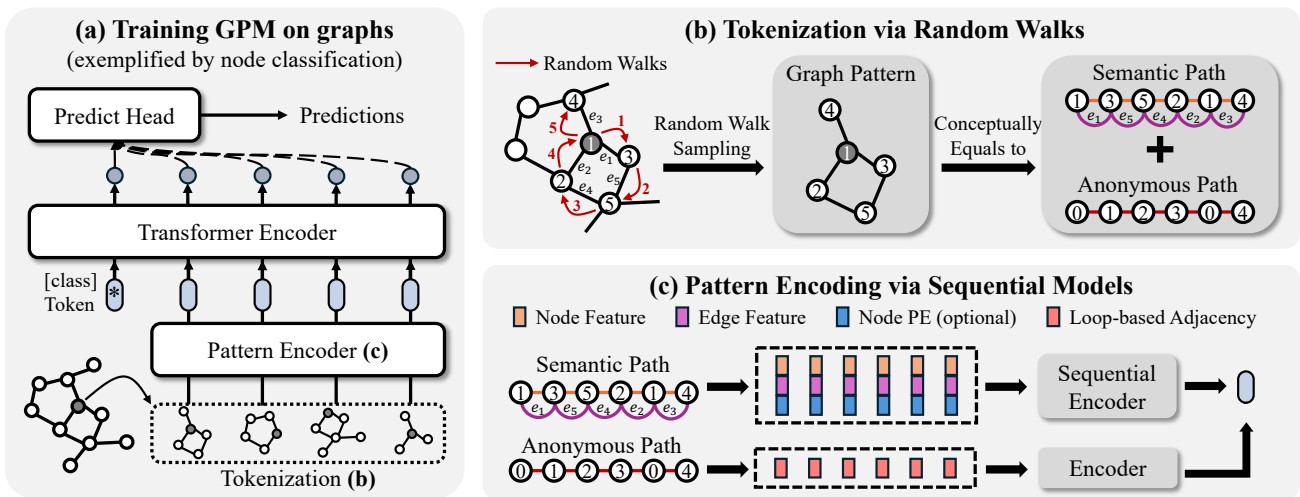

Figure 2. The overview framework of GPM.

et al., 2019; Fan et al., 2022; Jin et al., 2022; Wang & Cho, 2024) improve long-range modeling (Welke et al., 2023), but often sacrifice local pattern understanding (Tönshoff et al., 2023). Third, graph transformers (Kreuzer et al., 2021; Ying et al., 2021) utilize global attention to capture arbitrary node dependencies and exceed WL expressiveness, but incur quadratic complexity, limiting scalability to large graphs (Wu et al., 2023). While these approaches extend message passing in expressiveness, they are often implemented as extensions or complements to the message passing, failing to directly and effectively model graph patterns. This observation motivates the need for a fundamentally different approach, as pursued in this work. A more detailed discussion is provided in Appendix A.

**Graph Patterns as Tokens.** Early methods for leveraging graph patterns often relied on primitive training paradigms. For instance, DGK (Yanardag & Vishwanathan, 2015) uses graph kernels to measure relationships between (substructure) patterns, while AWE (Ivanov & Burnaev, 2018) employs anonymous random walks to encode pattern distributions in graphs. More recent approaches tokenize graphs into sequences of substructures. GraphViT (He et al., 2023), for example, splits a graph into multiple subgraphs using graph partitioning algorithms (Karypis, 1997), encodes each subgraph individually via message passing, and aggregates the resulting embeddings to represent the entire graph. Similarly, GMT (Baek et al., 2021) decomposes nodes into a multiset, with each set representing a specific graph pattern. However, these methods are limited to graph-level tasks. On the other hand, NAGphormer (Chen et al., 2023) and GC-Former (Chen et al., 2024) utilize Hop2token and neighborhood sampling, respectively, to tokenize patterns representing individual nodes; Yet, they are tailored for node-level tasks. To address this, SAT (Chen et al., 2022), GNN-AK

(Zhao et al., 2022), and GraphMamba (Behrouz & Hashemi, 2024) propose task-agnostic tokenization methods. However, these methods often rely on message passing either as the pattern encoder or as an auxiliary module, inheriting the limitations of message passing. While the aforementioned methods have shown empirical success, none fully meet our criteria: (1) a universal graph tokenizer, (2) an effective pattern encoder, and (3) the ability to identify important patterns relevant to downstream tasks. Our proposed GPM overcomes these challenges.

## 3. Neural Graph Pattern Machine

Let $\mathcal{G} = (\mathcal{V}, \mathcal{E}, \boldsymbol{X}, \boldsymbol{E})$ denote a graph, where $\mathcal{V}$ is the set of nodes with $|\mathcal{V}| = N$, $\mathcal{E} \subseteq \mathcal{V} \times \mathcal{V}$ is the set of edges with $|\mathcal{E}| = E$, and $\boldsymbol{X}$ and $\boldsymbol{E}$ represent the node and edge feature matrices, respectively. Each node $v \in \mathcal{V}$ is associated with a feature vector $\mathbf{x}_v \in \mathbb{R}^{d_n}$, and each edge $e \in \mathcal{E}$ is associated with a feature vector $\mathbf{e}_e \in \mathbb{R}^{d_e}$ (if applicable). We define graph patterns as subgraphs that represent small, recurring substructures, such as triangles, stars, cycles, etc.

### 3.1. Pattern Tokenizer

Tokenization is an essential step for converting a given instance into a sequence of patterns. Existing methods typically rely on pre-defined strategies. For example, large language models (LLMs) employ pre-defined vocabularies (Brown et al., 2020) to decompose a sentence into a sequence of tokens, where each token represents a concept or entity. Similarly, vision transformers use patch tokenizers to partition an image into a sequence of grids (Dosovitskiy et al., 2021). However, a universal tokenizer for graphs has yet to be established. A straightforward approach is to construct a unified substructure vocabulary via efficient sub-

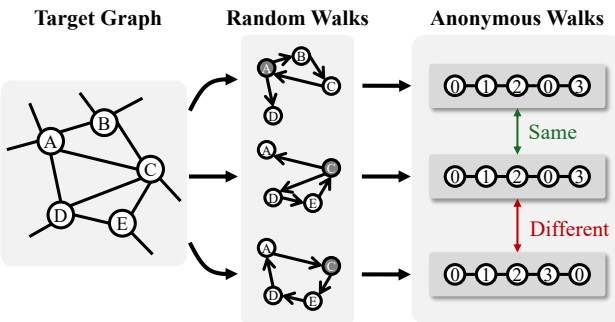

**Target Graph**     **Random Walks**     **Anonymous Walks**

*Figure 3.* Examples of anonymous paths.

graph matching (Sun et al., 2012), and then use this vocabulary to describe patterns for each instance (e.g., node, edge, or graph). However, the vocabulary construction and pattern matching are obviously inefficient and cannot scale well to large graphs. To address this challenge, GPM bypasses the need for an explicit fixed vocabulary by approximating the pattern matching process via random sampling.

The task now is to sample patterns that effectively describe graph instances. Ideally, the sampling process should be efficient and produce diverse patterns to enable scalability for large graphs. To achieve this, we propose leveraging random walks for sampling graph patterns. An (unbiased) random walk $w$ (length = $L$) is defined as a node sequence $w = (v_0, v_1, \ldots, v_L)$, generated via a Markov chain:

$$P(v_{i+1} \mid v_0, \ldots, v_i) = \mathbb{1}[(v_i, v_{i+1}) \in \mathcal{E}]/D(v_i), \quad (1)$$

where $D(v_i)$ represents the degree of node $v_i$. Utilizing this landing probability, it becomes straightforward to generate a sequence of random walks starting at node $v$.

Next, we discuss why this basic strategy is effective for sampling graph patterns and demonstrate that each random walk inherently depicts a unique graph pattern. Before delving into this, we introduce the concept of anonymous walks (Ivanov & Burnaev, 2018).

**Definition 3.1 (Anonymous Walk (Ivanov & Burnaev, 2018)).** Given a random walk $w = (v_0, v_1, \ldots, v_L)$, its corresponding anonymous walk is defined as a sequence of integers $\phi = (\gamma_0, \gamma_1, \ldots, \gamma_L)$, where $\gamma_i = \min pos(w, v_i)$. The mapping from the random walk to its anonymous path is represented by $w \mapsto \phi$.

Anonymous walks encode random walks as sequences of relative positions, preserving anonymity by avoiding the explicit recording of specific nodes. Each anonymous walk thus captures a unique graph topology pattern. For instance, as illustrated in Figure 3, the random walks "A-B-C-A-D" and "C-D-E-C-A" both correspond to the same anonymous path "0-1-2-0-3," representing a triangle-shaped substructure with an additional connection. In addition, the random

walk "A-C-E-D-A" maps to the anonymous path "0-1-2-3-0," which represents a rectangle-shaped substructure. Note we refer the original random walks as semantic paths.

Recent findings (Micali & Zhu, 2016; Ivanov & Burnaev, 2018) have demonstrated that the distribution of anonymous paths originating from a node $v$ is sufficient to reconstruct the subgraphs induced by all nodes within a fixed distance from $v$. Consequently, we state the following proposition, establishing that anonymous paths encapsulate sufficient topological information to describe each node.

**Proposition 3.2.** *(Informal) Given a node $v \in \mathcal{V}$, assume the task requires information from the $k$-hop ego-graph $\mathcal{B}(v, k) = (\mathcal{V}', \mathcal{E}')$. A sufficiently large set of patterns, sampled via $l$-length anonymous walks with $l = O(|\mathcal{E}'|)$, can provide distinguishable topological representations.*

The proof is sketched in Appendix C.1. This proposition demonstrates that the distribution of anonymous paths starting from a node is sufficient to capture its topological properties. Naturally, two nodes $u$ and $v$ can be considered to share similar graph patterns if their anonymous path distributions are similar. This proposition can be extended to links and graphs by treating them as combinations of nodes, with the corresponding distributions. Thus, for each random walk, we can derive both a semantic path (capturing node features) and an anonymous path (capturing topological structures), where their combination conceptually represents a certain graph pattern. Then, we have the proposition.

**Proposition 3.3.** *(Informal) Jointly encoding semantic paths and anonymous paths via any bijective mappings provides a comprehensive representation of the graph inductive biases encapsulated in the corresponding graph pattern.*

This proposition, proved in Appendix C.2, implies that graph patterns can be effectively encoded by combining the individually encoded semantic and anonymous paths.

**Applicability to Graph-based Tasks.** The method is capable of tokenizing nodes, edges, and graphs. For node-level tasks, the tokenizer samples $k$ patterns for a node $v$. For edge-level tasks, the tokenizer samples $k$ patterns starting from the endpoints $u$ and $v$ of an edge $e = (u, v)$. For graph-level tasks, the tokenizer samples $k$ patterns for each graph by randomly selecting starting nodes within the graph.

### 3.2. Pattern Encoder

As discussed above, any graph pattern can be represented as the combination of a semantic path and an anonymous path. Given a graph pattern, let the semantic path be $w = (v_0, \ldots, v_n)$ and the anonymous path be $\phi = (\gamma_0, \ldots, \gamma_n)$. These paths are encoded separately using two distinct encoders, which are combined to form the pattern embedding:

$$\boldsymbol{p} = \rho_s(w) + \lambda \cdot \rho_a(\phi), \quad (2)$$

where $p$ denotes the pattern embedding, $\rho_s$ and $\rho_a$ are the encoders for the semantic path and the anonymous path, respectively, and $\lambda$ is a weighting coefficient.

**Semantic Path Encoder.** To encode the semantic path, we construct a sequence of node features according to the semantic path as $[x_0, \ldots, x_n]$, where nodes may appear multiple times. The encoder $\rho_s$ can be any model capable of processing sequential data. By default, we use the transformer encoder due to its superior expressiveness in capturing long-range dependencies. The alternatives can be mean aggregator or GRU (Chung et al., 2014). The encoding process is defined as:

$$\rho_s(w) = \rho_s([h_0, \ldots, h_n]), \quad h_i = W x_i + b, \quad (3)$$

where $x = [x \parallel e]$ represents the concatenation of node features $x$ and optional edge features $e$ (if applicable). Note that in a path, the number of edges is one less than the number of nodes. To address this mismatch, edge features are padded with a zero vector at the beginning of the sequence.

**Node Positional Embedding.** To incorporate advanced topological information, it is optional to concatenate node positional embeddings (PEs) with the node features. This approach has been shown to be effective to enhance model expressiveness (Kreuzer et al., 2021; Rampasek et al., 2022; He et al., 2023; Chen et al., 2023). The augmented node features are represented as:

$$x_i = [x_i \parallel e_{i-1,i} \parallel \alpha_i], \quad (4)$$

where $x_i$ denotes the node features, $e_{i-1,i}$ represents optional edge features, and $\alpha_i$ refers to the optional positional embedding. In this work, we utilize widely adopted PEs, including random-walk structural embeddings (RWSE) (Rampasek et al., 2022) and Laplacian eigenvector embeddings (Lap) (Kreuzer et al., 2021). Empirically, we observe that the choice of PE depends on the dataset. Notably, the model can still achieve competitive performance even without positional embeddings.

**Anonymous Path Encoder.** For an anonymous path $\phi = (\gamma_0, \ldots, \gamma_n)$, we adopt a similar approach to encode the path as used for semantic paths, with a key distinction: anonymous nodes lack explicit features. Instead of employing one-hot encoding for each anonymous index, inspired by Tönshoff et al. (2023), we utilize an advanced method that encodes both anonymous indices and connectivity information. Specifically, for an anonymous path of length $k$, each node $v_i$ is assigned a $k$-dimensional vector $z_i$, where $z_{i,j} = \mathbb{1}[\gamma_i = \gamma_j]$. This encoding not only captures identity information but also encodes loop structures. We refer to this representation as loop-based adjacency. Consequently, any anonymous path can be expressed as $\phi = [z_0, \ldots, z_n]$, which can then be processed using encoder $\rho_a$:

$$\rho_a(\phi) = \rho_a([z_0, \ldots, z_n]). \quad (5)$$

In this work, we adopt GRU as the default encoder due to its balance of expressiveness and computational efficiency.

### 3.3. Important Pattern Identifier

For each graph instance, we use a set of encoded patterns $P = [p_0, \ldots, p_m]$ to describe its characteristics. Since the patterns are randomly sampled from the graph, it is essential to identify the most relevant patterns for downstream tasks. To achieve this, we employ a transformer encoder to highlight the dominant patterns by learning their relative importance. The encoding process is defined as follows:

$$Q = P W_Q, \quad K = P W_K, \quad V = P W_V, \quad (6)$$

$$\text{Attn}(P) = \text{softmax}\left(\frac{Q K^\top}{\sqrt{d_{out}}}\right) V \in \mathbb{R}^{n \times d_{out}}, \quad (7)$$

$$P' = \text{FFN}(P + \text{Attn}(P)), \quad (8)$$

where $W_Q, W_K, W_V$ are trainable parameter matrices, $d_{out}$ is the dimension of the query matrix $Q$, and FFN represents a two-layer MLP. We utilize multi-head attention, which has proven effective in practice by concatenating multiple attention mechanisms. Additionally, multiple transformer layers can be stacked to further enhance the capacity.

Let $P' = [p'_0, \ldots, p'_m]$ denote the output of the final transformer layer. These outputs are aggregated for downstream predictions using an additional prediction head:

$$\hat{y} = \text{Head}\left(\frac{1}{m}\sum_{i=1}^{m} p'_i\right), \quad (9)$$

where Head is a linear prediction layer. A mean readout function is applied over all pattern embeddings to compute the instance embedding, which is then used for prediction.

**Class Token.** For improved interpretability, a class token $p_{cls}$ can be appended to the sequence of pattern embeddings before transformer encoding, such that $P = [p_{cls}, p_0, \ldots, p_m]$. The downstream prediction is then based solely on the encoded class token, modifying Equation 9 as $\hat{y} = \text{Head}(p'_{cls})$. This approach evaluates the significance of individual graph patterns for downstream tasks based on the attention scores on the class token. Note that the primary purpose of using the class token is to enhance interpretability rather than to improve model performance, although it may also serve as an efficient summarization of the entire pattern set (Dosovitskiy et al., 2021).

### 3.4. Training Strategies

**Test-Time Augmentation.** For a single instance, we select $k$ patterns to represent its characteristics, where empirical evidence suggests that a larger $k$ generally improves performance. However, increasing $k$ imposes additional computational costs, as the self-attention has a quadratic

time complexity with respect to the input length, $O(k^2)$. Inspired by test-time scaling in LLMs (Snell et al., 2024), we address this trade-off by training the model with $m$ patterns and using $k$ patterns during inference, where $k \gg m$. By default, we set $m = 16$ and $k = 128$. To further reduce the computational overhead of pattern sampling, we pre-sample $k$ patterns during preprocessing and randomly select $m$ patterns during training. The time consumption is minimal, taking less than 10 seconds on most datasets. Even for the largest dataset used ($\sim$2,500,000 nodes), the sampling time remains under 2 minutes on an Nvidia A40 GPU.

**Multi-Scale Learning.** Recent studies have demonstrated the effectiveness of multi-scale learning across various domains (Liao et al., 2019; Chen et al., 2021). Inspired by the success of multi-scale learning in visual transformers (Chen et al., 2021), which process small and large patch tokens within a single model, we propose sampling patterns of varying sizes. Specifically, during tokenization, we sample patterns with lengths $[l_1, l_2, \dots]$ instead of using a fixed length $l$. By default, the multi-scale lengths are set to $[2, 4, 6, 8]$. Note the number of patterns remains unchanged. Additional implementation details are in Appendix B.

### 3.5. How Does GPM Surpass Message Passing?

**Advancing Expressivity.** Message passing frameworks can distinguish non-isomorphic graphs that are distinguishable by the 1-WL isomorphism test (Xu et al., 2019). We demonstrate that GPM surpasses message passing in expressiveness under the following mild assumptions: (1) the graphs are connected, unweighted, and undirected, and (2) the number of sampled patterns is sufficiently large.

**Theorem 3.4.** *Under the reconstruction conjecture assumption, GPM can distinguish all pairs of non-isomorphic graphs given a sufficient number of graph patterns.*

The proof is sketched by first demonstrating the expressive power of a simplified variant of GPM and then generalizing to the full model. The detailed proof is in Appendix C.3. Based on the theorem, it is readily to extend to the following corollary.

**Corollary 3.5.** *For $k \geq 1$, there exist graphs that can be distinguished by GPM using walk length $k$ but not by the $k$-WL isomorphism test.*

The proof of this result is detailed in Appendix C.4. This corollary highlights that GPM is at least as expressive as message passing frameworks and high-order GNNs. Empirically, GPM successfully distinguishes graphs that remain indistinguishable under the 3-WL test (Appendix E.1).

**Tackling Over-Squashing.** Another limitation of message passing is their focus on localized information, which prevents them from effectively capturing long-range dependencies within graphs. In contrast, GPM demonstrates superior capability in modeling long-range interactions. Following He et al. (2023), we evaluate this capability using the TREENEIGHBORSMATCHING (Alon & Yahav, 2021). Our GPM perfectly fits the data with task radius up to 7, yet message passing methods exhibit over-squashing effects as early as task radius 4 (Appendix E.2).

## 4. Experiments

### 4.1. Applicability to Graph-based Tasks

To evaluate the effectiveness of our method on all graph predictive tasks (node, link, and graph), we conduct extensive experiments such as node classification, link prediction, graph classification, and graph regression.

**Node Classification.** We conduct experiments on benchmark datasets of varying scales, with their statistics and homophily ratios summarized in Table 1. The datasets include Products, Computer, Arxiv, WikiCS, CoraFull, Deezer, Blog, Flickr, and Flickr-S (Small). We adopt the dataset splits from Chen et al. (2023) and Chen et al. (2024): public splits for WikiCS, Flickr, Arxiv, and Products; 60/20/20 train/val/test split for CoraFull and Computer; 50/25/25 split for the remaining datasets. Accuracy is used as the evaluation metric. The baselines include message passing GNNs (GCN, GAT, APPNP, GPR-GNN, OrderedGNN), random walk-based GNNs (RAW-GNN, RUM), and graph transformers (GraphGPS, SAN, NodeFormer, GOAT, NAGphormer, GraphMamba, VCR-Graphormer, and GCFormer). As presented in Table 1, our GPM consistently outperforms message passing GNNs and random walk-based GNNs across all datasets. When compared to graph transformers specifically designed for node classification, our method achieves superior performance on most datasets, with the exception of CoraFull. Although NAGphormer slightly outperforms GPM on CoraFull, its Hop2Token mechanism constraints the method on node-level tasks.

**Link Prediction.** We evaluate the link prediction performance on three datasets: Cora, Pubmed, and ogbl-Collab. Following Guo et al. (2023), we split the edges into 80/5/15 train/val/test sets and use Hits@20 for Cora and Pubmed, and Hits@50 for ogbl-Collab for evaluation. The baselines include GCN, LLP, RUM, NodeFormer, and NAGphormer. The results, along with dataset statistics, are summarized in Table 2. Notably, graph transformer methods such as NodeFormer and NAGphormer do not achieve competitive performance and, in some cases, perform worse than message passing GNNs. This discrepancy may arise from the inductive bias provided by message passing, which iteratively updates node representations based on their neighborhoods and is inherently well-suited for modeling connectivity between nodes. Our GPM achieves the best performance across all evaluated datasets. This superior performance is

*Table 1.* Node classification results, where the best-performing model is highlighted in **bold**, and the second-best method is underlined.

|  | PRODUCTS | COMPUTER | ARXIV | WIKICS | CORAFULL | DEEZER | BLOG | FLICKR | FLICKR-S |
|---|---|---|---|---|---|---|---|---|---|
| **# Nodes** | 2,449,029 | 13,752 | 169,343 | 11,701 | 19,793 | 28,281 | 5,196 | 89,250 | 7,575 |
| **# Edges** | 123,718,024 | 491,722 | 2,315,598 | 431,206 | 126,842 | 185,504 | 343,486 | 899,756 | 479,476 |
| **Homophily Ratio H($\mathcal{G}$)** | 0.81 | 0.78 | 0.65 | 0.65 | 0.57 | 0.53 | 0.40 | 0.32 | 0.24 |
| GCN (Kipf & Welling, 2017) | $75.64_{\pm0.21}$ | $89.65_{\pm0.52}$ | $71.74_{\pm0.29}$ | $77.47_{\pm0.85}$ | $61.76_{\pm0.14}$ | $62.70_{\pm0.70}$ | $94.12_{\pm0.79}$ | $50.90_{\pm0.12}$ | $84.58_{\pm0.49}$ |
| GAT (Veličković et al., 2018) | $79.45_{\pm0.59}$ | $90.78_{\pm0.13}$ | $72.01_{\pm0.20}$ | $76.91_{\pm0.82}$ | $64.47_{\pm0.18}$ | $61.70_{\pm0.80}$ | $93.47_{\pm0.63}$ | $50.70_{\pm0.32}$ | $85.11_{\pm0.67}$ |
| APPNP (Gasteiger et al., 2019) | $77.58_{\pm0.47}$ | $90.18_{\pm0.17}$ | $69.40_{\pm0.50}$ | $78.87_{\pm0.11}$ | $65.16_{\pm0.28}$ | $66.10_{\pm0.60}$ | $94.77_{\pm0.19}$ | $50.36_{\pm0.55}$ | $84.66_{\pm0.31}$ |
| GPRGNN (Chien et al., 2021) | $79.76_{\pm0.59}$ | $89.32_{\pm0.29}$ | $71.10_{\pm0.12}$ | $78.12_{\pm0.23}$ | $67.12_{\pm0.31}$ | $63.20_{\pm0.84}$ | $94.36_{\pm0.29}$ | $48.32_{\pm1.20}$ | $85.91_{\pm0.51}$ |
| OrderedGNN (Song et al., 2023) | - | $\underline{92.03_{\pm0.13}}$ | - | $\underline{79.01_{\pm0.68}}$ | $69.21_{\pm0.23}$ | $66.12_{\pm0.75}$ | $95.90_{\pm0.44}$ | $51.20_{\pm0.32}$ | $\underline{88.68_{\pm0.54}}$ |
| RAW-GNN (Jin et al., 2022) | - | $90.98_{\pm0.73}$ | - | $78.01_{\pm0.58}$ | $68.64_{\pm0.55}$ | $65.11_{\pm0.64}$ | $94.96_{\pm0.70}$ | $49.58_{\pm0.38}$ | $86.53_{\pm0.65}$ |
| RUM (Wang & Cho, 2024) | $78.68_{\pm1.40}$ | $90.62_{\pm0.24}$ | $70.54_{\pm0.30}$ | $78.20_{\pm0.29}$ | $70.42_{\pm0.08}$ | $64.25_{\pm0.62}$ | $94.16_{\pm0.35}$ | $50.97_{\pm0.32}$ | $87.25_{\pm0.66}$ |
| GraphGPS (Rampasek et al., 2022) | OOM | $91.19_{\pm0.54}$ | $70.97_{\pm0.41}$ | $78.66_{\pm0.49}$ | $55.76_{\pm0.23}$ | $60.56_{\pm0.62}$ | $94.35_{\pm0.52}$ | $45.15_{\pm2.41}$ | $83.61_{\pm0.70}$ |
| SAN (Kreuzer et al., 2021) | - | $89.83_{\pm0.16}$ | - | $78.46_{\pm0.99}$ | $59.01_{\pm0.34}$ | $64.29_{\pm0.35}$ | $90.21_{\pm0.20}$ | OOM | OOM |
| NodeFormer (Wu et al., 2022) | $72.93_{\pm0.13}$ | $86.98_{\pm0.62}$ | $67.19_{\pm0.83}$ | $74.73_{\pm0.94}$ | $71.01_{\pm0.14}$ | $66.40_{\pm0.70}$ | $93.79_{\pm0.76}$ | $\underline{51.23_{\pm0.64}}$ | $88.30_{\pm0.22}$ |
| GOAT (Kong et al., 2023) | $\underline{82.00_{\pm0.43}}$ | $90.96_{\pm0.90}$ | $72.41_{\pm0.40}$ | $77.00_{\pm0.77}$ | $68.55_{\pm0.34}$ | $65.31_{\pm0.24}$ | $94.40_{\pm0.08}$ | $48.30_{\pm0.47}$ | $88.16_{\pm0.95}$ |
| NAGphormer (Chen et al., 2023) | $73.55_{\pm0.21}$ | $91.22_{\pm0.14}$ | $70.13_{\pm0.55}$ | $77.16_{\pm0.72}$ | $\textbf{71.51}_{\pm0.13}$ | $65.54_{\pm0.57}$ | $94.42_{\pm0.63}$ | $49.66_{\pm0.29}$ | $86.85_{\pm0.85}$ |
| GraphMamba (Behrouz & Hashemi, 2024) | - | - | $72.48_{\pm0.00}$ | - | - | - | - | - | - |
| VCR-Graphormer (Fu et al., 2024) | - | $91.04_{\pm0.12}$ | - | $77.69_{\pm0.33}$ | $68.78_{\pm0.29}$ | $65.28_{\pm0.51}$ | $93.92_{\pm0.37}$ | $50.77_{\pm0.61}$ | $86.23_{\pm0.74}$ |
| GCFormer (Chen et al., 2024) | - | $91.63_{\pm0.18}$ | - | $78.12_{\pm0.50}$ | $69.70_{\pm0.54}$ | $65.16_{\pm0.33}$ | $\underline{96.03_{\pm0.44}}$ | $50.28_{\pm0.69}$ | $87.90_{\pm0.45}$ |
| GPM | $\textbf{82.62}_{\pm0.39}$ | $\textbf{92.28}_{\pm0.39}$ | $\textbf{72.89}_{\pm0.68}$ | $\textbf{80.19}_{\pm0.41}$ | $\underline{71.23_{\pm0.51}}$ | $\textbf{67.26}_{\pm0.22}$ | $\textbf{96.71}_{\pm0.59}$ | $\textbf{52.22}_{\pm0.19}$ | $\textbf{89.41}_{\pm0.47}$ |

*Table 2.* Link prediction results.

|  | CORA | PUBMED | OGBL-COLLAB |
|---|---|---|---|
| **# Nodes** | 2,708 | 19,717 | 235,868 |
| **# Edges** | 10,556 | 88,648 | 2,570,930 |
| GCN (Kipf & Welling, 2017) | $84.14_{\pm1.19}$ | $85.06_{\pm3.79}$ | $44.75_{\pm1.07}$ |
| LLP (Guo et al., 2023) | $\underline{89.95_{\pm2.01}}$ | $\underline{87.23_{\pm4.92}}$ | $\underline{49.10_{\pm0.57}}$ |
| RUM (Wang & Cho, 2024) | $88.74_{\pm0.60}$ | $85.87_{\pm3.93}$ | $48.19_{\pm0.94}$ |
| NodeFormer (Wu et al., 2022) | $80.78_{\pm1.44}$ | $83.93_{\pm4.72}$ | $46.56_{\pm0.62}$ |
| NAGphormer (Chen et al., 2023) | $86.87_{\pm1.58}$ | $86.46_{\pm4.09}$ | $47.56_{\pm0.52}$ |
| GPM | $\textbf{92.85}_{\pm0.54}$ | $\textbf{88.29}_{\pm5.15}$ | $\textbf{49.70}_{\pm0.59}$ |

likely due to the effectiveness of the captured patterns in accurately reflecting the connectivity between nodes.

**Graph Classification and Regression.** We evaluate the model on six graph datasets: social networks (IMDB-B, COLLAB, Reddit-M5K, Reddit-M12K) for classification and molecule graphs (ZINC and ZINC-Full) for regression. We use 80/10/10 train/val/test splits for social networks, and the public splits for molecule graphs. The baselines include message passing GNNs (GIN, PNA, GNN-AK), graph kernels (DGK), random walk-based methods (AWE, CRaWl, AgentNet, RUM), and graph transformers (GMT, SAN, Graphormer, GPS, SAT, DeepGraph, GraphViT, GEANet). The results, along with dataset statistics, are presented in Table 3. We observe that graph transformers generally outperform message passing GNNs, likely due to their superior capability in modeling long-range dependencies. Notably, our GPM consistently outperforms all other methods across datasets of varying scales, particularly outperforming methods that also utilize graph patterns as tokens (e.g., GMT, SAT, GraphViT). This might because these methods still rely on message passing as the encoder for individual patterns, thereby inheriting the limitations of message passing. In

contrast, our GPM eliminates the need for message passing, potentially enabling more effective substructure learning.

### 4.2. Out-of-Distribution Generalization

We evaluate the model robustness under distribution shifts between training and testing sets. Under the setting, the model is trained on a source graph and evaluated on a target graph, with a 20/80 val/test split. We conduct experiments on citation networks, ACM and DBLP (using accuracy as the metric), as well as social networks Twitch (using AUC as the metric). The Twitch dataset consists of six graphs (DE, EN, ES, FR, PT, RU), where the model is trained on DE and evaluated on the remaining graphs. For baselines, in addition to the methods used in previous experiments, we include domain adaptation and OOD generalization baselines such as DANN, SR-GNN, StruRW, and SSReg. The experimental results are summarized in Table 4. We observe that standard graph learning methods struggle in this setting, highlighting their limited robustness to OOD testing. In contrast, our GPM outperforms existing OOD-specific methods in settings such as A → D and Twitch, demonstrating superior robustness to OOD challenges. This can be attributed to GPM's pattern learning ability that potentially identifies shared patterns between source and target graphs, whereas message passing is sensitive to subtle structural changes (Wang et al., 2024c). Furthermore, the performance of GPM is enhanced when combined with OOD techniques such as DANN and SSReg, achieving significant improvements across all settings, particularly in D → A.

### 4.3. Scalability

**Large Graphs.** Each graph instance (e.g., node, edge, or graph) is represented by $k$ patterns, with an encoding com-

*Table 3.* Graph classification and regression (i.e., ZINC and ZINC-Full) results.

| | IMDB-B ↑ | COLLAB ↑ | REDDIT-M5K ↑ | REDDIT-M12K ↑ | ZINC ↓ | ZINC-FULL ↓ |
|---|---|---|---|---|---|---|
| **# Graphs** | 1,000 | 5,000 | 4,999 | 11,929 | 12,000 | 249,456 |
| **# Nodes (in average)** | ∼19.8 | ∼74.5 | ∼508.5 | ∼391.4 | ∼23.2 | ∼23.2 |
| **# Edges (in average)** | ∼193.1 | ∼4914.4 | ∼1189.7 | ∼913.8 | ∼49.8 | ∼49.8 |
| GIN (Xu et al., 2019) | 73.26$_{\pm0.46}$ | 80.59$_{\pm0.27}$ | 45.88$_{\pm0.78}$ | 39.37$_{\pm1.40}$ | 0.526$_{\pm0.051}$ | 0.088$_{\pm0.002}$ |
| DGK (Yanardag & Vishwanathan, 2015) | 66.96$_{\pm0.56}$ | 73.09$_{\pm0.25}$ | 41.27$_{\pm0.18}$ | 32.22$_{\pm0.10}$ | - | - |
| PNA (Corso et al., 2020) | 72.31$_{\pm3.67}$ | 74.73$_{\pm1.09}$ | 42.18$_{\pm2.96}$ | 38.57$_{\pm1.86}$ | 0.142$_{\pm0.010}$ | 0.067$_{\pm0.009}$ |
| GNN-AK+ (Zhao et al., 2022) | 75.00$_{\pm4.20}$ | 77.35$_{\pm0.93}$ | 47.78$_{\pm1.12}$ | 40.60$_{\pm0.99}$ | 0.080$_{\pm0.001}$ | 0.034$_{\pm0.007}$ |
| AWE (Ivanov & Burnaev, 2018) | 74.45$_{\pm5.83}$ | 73.93$_{\pm1.94}$ | 50.46$_{\pm1.91}$ | 39.20$_{\pm2.09}$ | 0.094$_{\pm0.005}$ | 0.059$_{\pm0.005}$ |
| CRaWl (Tönshoff et al., 2023) | 73.69$_{\pm2.05}$ | 77.17$_{\pm0.78}$ | 48.81$_{\pm1.67}$ | 40.72$_{\pm0.65}$ | 0.085$_{\pm0.004}$ | 0.036$_{\pm0.005}$ |
| AgentNet (Martinkus et al., 2023) | 75.88$_{\pm3.60}$ | 77.30$_{\pm1.98}$ | 47.71$_{\pm0.94}$ | 42.15$_{\pm0.13}$ | 0.144$_{\pm0.016}$ | 0.040$_{\pm0.006}$ |
| RUM (Wang & Cho, 2024) | 81.10$_{\pm4.50}$ | 75.50$_{\pm0.58}$ | 48.66$_{\pm0.76}$ | 41.66$_{\pm0.15}$ | - | - |
| GMT (Baek et al., 2021) | 73.48$_{\pm0.76}$ | 78.94$_{\pm0.44}$ | 49.96$_{\pm1.21}$ | 40.63$_{\pm0.94}$ | - | - |
| SAN (Kreuzer et al., 2021) | 76.00$_{\pm1.90}$ | 74.45$_{\pm2.46}$ | 50.76$_{\pm0.41}$ | 39.92$_{\pm1.02}$ | 0.139$_{\pm0.006}$ | - |
| Graphormer (Ying et al., 2021) | 76.74$_{\pm0.86}$ | 78.82$_{\pm1.21}$ | 48.98$_{\pm0.30}$ | 41.42$_{\pm0.42}$ | 0.122$_{\pm0.006}$ | 0.025$_{\pm0.004}$ |
| GPS (Rampasek et al., 2022) | 77.76$_{\pm0.98}$ | 77.41$_{\pm0.56}$ | 49.09$_{\pm0.71}$ | 41.55$_{\pm0.16}$ | 0.070$_{\pm0.004}$ | - |
| SAT (Chen et al., 2022) | 78.29$_{\pm1.26}$ | 78.35$_{\pm0.85}$ | 47.02$_{\pm0.88}$ | 42.14$_{\pm0.07}$ | 0.094$_{\pm0.008}$ | 0.036$_{\pm0.002}$ |
| DeepGraph (Zhao et al., 2023) | - | - | - | - | 0.072$_{\pm0.004}$ | - |
| GraphViT (He et al., 2023) | 78.05$_{\pm1.00}$ | 78.79$_{\pm0.74}$ | 48.39$_{\pm0.78}$ | 40.17$_{\pm0.52}$ | 0.073$_{\pm0.001}$ | 0.035$_{\pm0.005}$ |
| GEANet (Liang et al., 2024) | - | - | - | - | 0.193$_{\pm0.001}$ | - |
| GPM | **82.67$_{\pm0.47}$** | **80.70$_{\pm0.74}$** | **51.87$_{\pm1.04}$** | **43.07$_{\pm0.29}$** | **0.064$_{\pm0.004}$** | **0.021$_{\pm0.002}$** |

*Table 4.* Out-of-distribution (OOD) generalization results.

| | A → D | D → A | TWITCH |
|---|---|---|---|
| GCN (Kipf & Welling, 2017) | 59.02$_{\pm1.04}$ | 54.26$_{\pm0.78}$ | 56.68 |
| GAT (Veličković et al., 2018) | 61.67$_{\pm3.54}$ | 58.98$_{\pm5.78}$ | 56.86 |
| APPNP (Gasteiger et al., 2019) | 49.05$_{\pm8.94}$ | 57.54$_{\pm4.32}$ | 54.26 |
| RUM (Wang & Cho, 2024) | 60.38$_{\pm3.69}$ | 52.95$_{\pm6.91}$ | 58.59 |
| NodeFormer (Wu et al., 2022) | 65.69$_{\pm5.09}$ | 55.33$_{\pm2.25}$ | 54.17 |
| NAGphormer (Chen et al., 2023) | 65.01$_{\pm4.10}$ | 55.40$_{\pm2.31}$ | 54.19 |
| DANN (Ganin et al., 2016) | 68.92$_{\pm3.29}$ | 63.07$_{\pm1.65}$ | 60.71 |
| SR-GNN (Zhu et al., 2021) | 68.30$_{\pm1.26}$ | 62.43$_{\pm1.08}$ | 59.65 |
| StruRW (Liu et al., 2023) | 70.19$_{\pm2.10}$ | 65.07$_{\pm1.98}$ | 61.46 |
| SSReg (You et al., 2023) | 69.04$_{\pm2.95}$ | 65.93$_{\pm1.05}$ | 60.43 |
| GPM | 74.91$_{\pm5.07}$ | 59.57$_{\pm2.97}$ | 61.63 |
| GPM + DANN | 75.39$_{\pm3.98}$ | 63.03$_{\pm3.58}$ | **62.98** |
| GPM + SSReg | **75.66$_{\pm3.04}$** | 67.30$_{\pm1.94}$ | 62.77 |

A and D are abbreviation of ACM and DBLP. Twitch is averaged over 5 settings.

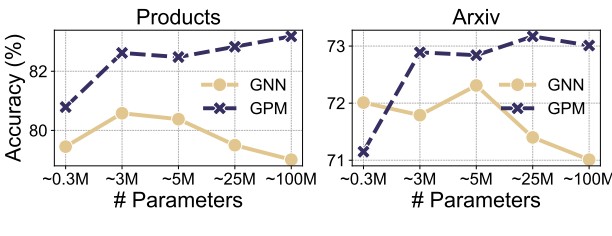

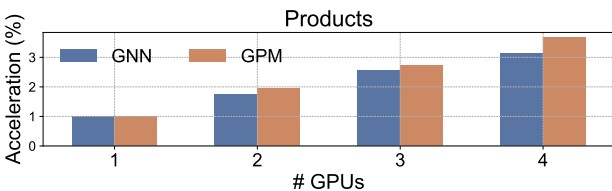

*Figure 4.* Model scalability analysis. (Top) Number of Parameters vs. Accuracy. (Bottom) Number of GPUs vs. Acceleration Ratio.

plexity of $O(k^2)$ and an overall complexity of $O(n \cdot k^2)$ (see Appendix D), where $n \gg k^2$ denotes the number of instances. This design enables GPM to efficiently scale to large graphs using mini-batch training. We evaluate GPM on large-scale graph datasets, e.g., Products, ogbl-Collab, and ZINC-Full (Tables 1, 2, and 3). GPM achieves competitive performance across these large-scale benchmarks.

**Large Models.** Leveraging the transformer architecture, GPM naturally scales to larger model sizes by stacking additional transformer layers, as illustrated in Figure 4 (Top). Empirically, increasing model parameters enhances performance on large-scale graphs. In contrast, message passing GNNs (GAT in this case) struggle to scale due to the over-smoothing effect. The architectural details of large-scale GPM models are presented in Table 7 in Appendix.

**Distributed Training.** Transformers have demonstrated remarkable efficiency in distributed training due to the parallelization of self-attention mechanism (Shoeybi et al., 2019). In contrast, message passing GNNs are less efficient for distributed training, as their iterative message passing introduces sequential dependencies and incurs significant communication overhead when computational nodes are distributed across machines. By leveraging a transformer-based architecture, GPM achieves superior efficiency in distributed training compared to message passing GNNs on PRODUCTS, as shown in Figure 4 (Bottom). Further details

*Table 5.* Model component ablation. "PE" denotes positional embedding, "AP" is anonymous path, and "SP" is semantic path.

| | | PRODUCTS | ARXIV | OGBL-COLLAB | COLLAB |
|---|---|---|---|---|---|
| Comp. | GPM | **82.62**±0.39 | **72.89**±0.68 | **49.70**±0.59 | **80.70**±0.74 |
| | w/o PE | 82.47±0.21 | 72.59±0.23 | 49.59±0.65 | 78.00±1.07 |
| | w/o AP | 81.99±0.68 | 72.47±0.40 | 48.43±0.97 | 78.33±0.24 |
| | w/o PE & AP | 80.74±0.65 | 71.40±0.76 | 48.24±0.97 | 75.40±0.99 |
| SP Enc. | Mean | 80.42±0.48 | 72.08±0.66 | 47.83±1.00 | 76.27±0.47 |
| | GRU | 80.91±0.33 | 72.23±0.41 | 48.12±0.55 | 74.00±1.57 |
| | Transformer | **82.62**±0.39 | **72.89**±0.68 | **49.70**±0.59 | **80.70**±0.74 |

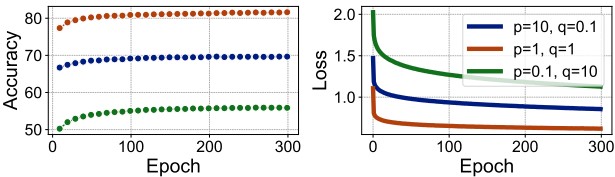

*Figure 5.* Training loss and model performance on PRODUCTS with varying sampling criteria.

and discussions are provided in Appendix F.1.

### 4.4. Ablation Study

**Impact of Model Components.** Table 5 presents the model component ablation study results. Both positional embeddings (PE) and anonymous paths (AP) contribute to capturing topological information, with PE encoding relative node positions and AP characterizing pattern structures. Empirically, AP has a greater impact than PE, suggesting that the model prioritizes understanding pattern structures over node locations. For the semantic path (SP) encoder, the Transformer achieves the best due to its ability to adaptively model both localized and long-range dependencies. More results and discussions are provided in Appendix F.2.

**Impact of Training Tricks.** Both multi-scale training and test-time augmentation contribute to improved performance. Specifically, multi-scale training increases average accuracy from 70.72 to 72.34 by incorporating hierarchical substructure knowledge. Test-time augmentation uses fewer substructure patterns during training (e.g., 16 at training vs. 128 at inference), which significantly reduces training cost with negligible performance drop (72.43 to 72.34). Full details are provided in Appendix F.3.

**GPM Automatically Learns Data Dependencies.** Leveraging the transformer architecture, GPM automatically identifies dominant patterns. Experiments on the PRODUCTS (Figure 5) demonstrate that unbiased random walk sampling ($p = 1, q = 1$) achieves the best results, over localized ($p = 0.1, q = 10$) and long-range ($p = 10, q = 0.1$) sampling, by allowing the model to autonomously balance localized and long-range dependencies. Detailed discussions are provided in Appendix G.

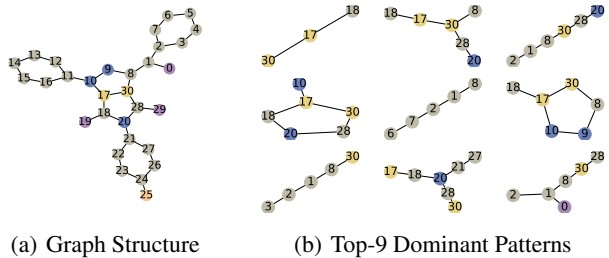

(a) Graph Structure      (b) Top-9 Dominant Patterns

*Figure 6.* Model interpretation on ZINC.

### 4.5. Model Interpretation

GPM leverages self-attention to identify the most relevant patterns for downstream tasks by utilizing a class token to aggregate pattern information. As an illustrative example, Figure 6 presents the 24-th molecule graph from ZINC (colors indicate different atom types) along with its top-9 key patterns, demonstrating that GPM effectively captures topologically significant structures such as stars and rings in molecules. Similarly, Figure 8 in Appendix visualizes the results on COMPUTERS, highlighting that triangle structures are predominant in e-commerce networks.

## 5. Conclusion

We propose GPM, a novel graph representation learning framework that directly learns from graph substructure patterns, eliminating the need for message passing. The architecture comprises three key components: a pattern sampler, a pattern encoder, and an important pattern identifier. Extensive experiments across node-, link-, and graph-level tasks demonstrate the effectiveness of GPM, showcasing its superior robustness, scalability, and interpretability. Moreover, GPM offers enhanced model expressiveness and a greater capacity for capturing long-range dependencies.

**Limitations and Future Works.** The prediction performance of GPM heavily depends on the quality of the sampled patterns. In this work, we adopt a random sampling strategy, aiming to sample as many patterns as possible to construct a comprehensive pattern set. However, this approach may increase resource consumption during model training and hyperparameter tuning. Developing an adaptive sampling strategy tailored to specific downstream tasks or designing a unified pattern vocabulary (Wang et al., 2024a;b) could mitigate this issue. Furthermore, the current implementation of GPM is limited to supervised tasks. Future extensions could include unsupervised learning (He et al., 2022), integration with LLMs (Yuan et al., 2021; Liu et al., 2024), incorporating external knowledge (Ni et al., 2025), or adaptations for complex graph types, such as heterogeneous graphs (Wang et al., 2023).

## Acknowledgments

This work was partially supported by the NSF under grants IIS-2321504, IIS-2334193, IIS-2340346, IIS-2217239, CNS-2426514, and CMMI-2146076, and Notre Dame Strategic Framework Research Grant (2025). Any opinions, findings, and conclusions or recommendations expressed in this material are those of the authors and do not necessarily reflect the views of the sponsors.

## Impact Statement

We introduce a novel graph representation learning framework that goes beyond the traditional message passing. By design, our GPM addresses inherent challenges in message passing, such as restricted expressiveness and over-squashing. This capability enables our framework to better scale to complex tasks requiring large models or involving large graphs. Also, GPM provides an interface to enable model interpretation, facilitating the applications requiring clear insights on graph topology, such as social network analysis and drug discovery. Potential ethical considerations include the misuse of the technology in sensitive domains, such as surveillance or profiling, and the reinforcement of biases present in the training data.

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

## A. Limitations of Message Passing and Recent Advances

The message passing paradigm (Kipf & Welling, 2017; Hamilton et al., 2017; Veličković et al., 2018) in GNNs has well-documented limitations, including restricted expressiveness, over-smoothing, over-squashing, and an inability to effectively model long-range dependencies. Given our focus on learning graph patterns, this discussion emphasizes the issues of expressiveness and notable advancements. Pioneering work by Xu et al. (2019) revealed that the expressive power of GNNs is fundamentally bounded by the 1-WL isomorphism test. Building on this, Corso et al. (2020) demonstrated that no GNN employing a single aggregation function can achieve the expressiveness of the 1-WL test when the neighborhood multiset has uncountable support. Consequently, this constraint renders GNNs incapable of identifying critical graph structures, such as stars, conjoint cycles, and $k$-cliques (Chen et al., 2020; Garg et al., 2020; Zhang et al., 2024a).

To address the expressiveness limitations of GNNs, three primary strategies have emerged. The first focuses on developing expressive GNNs that enhance the message passing framework to surpass the constraints of the 1-WL test. For instance, Maron et al. (2019b); Chen et al. (2019); Maron et al. (2019a) introduced $k$-order WL GNNs to emulate the $k$-WL test within GNN architectures. Similarly, Alsentzer et al. (2020); Bouritsas et al. (2022); Bodnar et al. (2021); Zhao et al. (2022) proposed advanced message passing mechanisms capable of detecting substructures in graphs, while Murphy et al. (2019); Loukas (2020) incorporated node positional embeddings to boost the representational power of message passing GNNs. Despite their increased expressiveness, these approaches often suffer from computational inefficiency (Azizian & marc lelarge, 2021).

An alternative approach leverages random walk kernels to guide the message passing process, constraining interactions to a limited range of nodes (Jin et al., 2022; Martinkus et al., 2023; Tönshoff et al., 2023; Wang & Cho, 2024; Chen et al., 2025). Notably, Tönshoff et al. (2023) demonstrated that random walk-based methods can capture both small substructures and long-range dependencies, while Wang & Cho (2024) showed that sufficiently long random walks can distinguish non-isomorphic graphs. Moreover, these random walk approaches are theoretically more expressive than message passing (Zhang et al., 2019; Fan et al., 2022; Welke et al., 2023; Michel et al., 2023; Graziani et al., 2024). Despite their strengths, these methods tend to emphasize long-range dependencies at the expense of localized information (Tönshoff et al., 2023) and lack interpretability regarding the specific graph knowledge being learned.

Lastly, graph transformers (GTs) (Kreuzer et al., 2021; Ying et al., 2021; Dwivedi & Bresson, 2020; Rampasek et al., 2022; He et al., 2023; Chen et al., 2022) have emerged as a compelling alternative to traditional message passing GNNs. Leveraging a global attention mechanism, GTs can capture correlations between any pair of nodes, enabling effective modeling of long-range dependencies. Both theoretical and empirical studies (Kreuzer et al., 2021; Ying et al., 2021) demonstrate that, under mild assumptions, graph transformers surpass the expressive power of WL isomorphism tests. This represents a fundamental advantage over message passing GNNs in terms of expressiveness. However, the quadratic complexity of all-pair node attention poses significant computational challenges, limiting the applicability of GTs to smaller graphs, such as molecular graphs (Wu et al., 2023).

Crucially, the aforementioned methods primarily focus on developing advanced message passing frameworks rather than directly encoding graph patterns. As a result, they may still inherit the limitations associated with message passing.

## B. Implementation Details

### B.1. Environments

Most experiments are conducted on Linux servers equipped with four Nvidia A40 GPUs. The models are implemented using PyTorch 2.4.0, PyTorch Geometric 2.6.1, and PyTorch Cluster 1.6.3, with CUDA 12.1 and Python 3.9.

### B.2. Training Details

The training of transformers is highly sensitive to regularization techniques. In our setup, we use the AdamW optimizer with weight decay and apply early stopping after 100 epochs. Label smoothing is set to 0.05, and gradient clipping is fixed at 1.0 to stabilize training. The learning rate follows a warm-up schedule with 100 warm-up steps by default.

All experiments are conducted five times with different random seeds. The batch size is set to 256 by default.

## B.3. Model Configurations

We perform hyperparameter search over the following ranges: learning rate $\{1e\text{-}2, 5e\text{-}3, 1e\text{-}3\}$, positional embedding dimension $\{4, 8, 20\}$, dropout $\{0.1, 0.3, 0.5\}$, weight decay $\{1e\text{-}2, 0\}$, and weighting coefficient $\lambda \in \{0.1, 0.5, 1.0\}$. For pattern sampling, we set $p = 1, q = 1$ by default (see Appendix G for details). The model configuration includes a hidden dimension of 256, 4 attention heads, and 1 transformer layer. The selected hyperparameters are summarized in Table 6.

*Table 6.* Hyper-parameter settings on predictive tasks.

| Task | PRODUCTS Node | COMPUTER Node | ARXIV Node | WIKICS Node | CORAFULL Node | DEEZER Node | BLOG Node | FLICKR Node | FLICKR-S Node |
|---|---|---|---|---|---|---|---|---|---|
| **Learning Rate** | 0.01 | 0.01 | 0.01 | 0.01 | 0.01 | 0.01 | 0.001 | 0.01 | 0.005 |
| **Dropout** | 0.3 | 0.1 | 0.1 | 0.5 | 0.1 | 0.1 | 0.1 | 0.5 | 0.3 |
| **Decay** | 0 | 0.01 | 0 | 0.01 | 0 | 0 | 0.01 | 0.01 | 0.01 |
| **Batch Size** | 256 | 256 | 256 | 256 | 256 | 256 | 256 | 256 | 256 |
| **PE Type** | Lap | Lap | Lap | Lap | Lap | Lap | Lap | Lap | Lap |
| **PE Dim** | 4 | 8 | 4 | 8 | 8 | 16 | 16 | 8 | 4 |
| **AP Encoder** | MEAN | GRU | GRU | GRU | GRU | GRU | GRU | MEAN | GRU |
| $\lambda$ | 0.5 | 0.5 | 1 | 1 | 1 | 0.5 | 0.1 | 0.5 | 0.5 |

| Task | CORA Link | PUBMED Link | OGBL-COLLAB Link | IMDB-B Graph | COLLAB Graph | REDDIT-M5K Graph | REDDIT-M12K Graph | ZINC Graph | ZINC-FULL Graph |
|---|---|---|---|---|---|---|---|---|---|
| **Learning Rate** | 0.001 | 0.01 | 0.01 | 0.001 | 0.01 | 0.001 | 0.005 | 0.01 | 0.01 |
| **Dropout** | 0.3 | 0.3 | 0.1 | 0.1 | 0.1 | 0.1 | 0.1 | 0.1 | 0.1 |
| **Decay** | 0 | 0.01 | 0 | 0 | 0 | 0 | 0.01 | 0 | 0 |
| **Batch Size** | 256 | 256 | 256 | 1024 | 1024 | 1024 | 256 | 1024 | 1024 |
| **PE Type** | Lap | Lap | Lap | RW | RW | RW | RW | RW | RW |
| **PE Dim** | 4 | 4 | 4 | 8 | 4 | 8 | 8 | 8 | 8 |
| **AP Encoder** | GRU | GRU | GRU | MEAN | MEAN | MEAN | GRU | GRU | GRU |
| $\lambda$ | 0.5 | 1 | 0.5 | 0.1 | 0.1 | 0.1 | 0.1 | 1 | 1 |

## B.4. Architectures in Model Scaling Analysis

In our scalability analysis (Section 4.3), we evaluate the performance of GAT and GPM across different model scales. The detailed model architectures and corresponding parameter counts are provided in Table 7.

*Table 7.* Model architectures in model scaling analysis.

| *Architectures of GNN (GAT in this case)* | | | | | |
|---|---|---|---|---|---|
| # GNN Layers | 2 | 2 | 2 | 3 | 3 |
| # Number of Heads | 8 | 24 | 32 | 48 | 112 |
| # Hidden Dimension | 512 | 1536 | 2048 | 3072 | 7168 |
| ARXIV | 0.35M | 2.63M | 4.56M | 19.44M | 104.07M |
| PRODUCTS | 0.34M | 2.6M | 4.52M | 19.37M | 103.92M |
| *Architectures of GPM* | | | | | |
| # Transformer Layers | 1 | 1 | 3 | 3 | 3 |
| # Number of Heads | 4 | 4 | 4 | 8 | 16 |
| # Hidden Dimension | 64 | 256 | 256 | 512 | 1024 |
| ARXIV | 0.19M | 2.94M | 4.52M | 20.04M | 96.72M |
| PRODUCTS | 0.21M | 3.21M | 4.79M | 21.12M | 100.96M |

# C. Proof

## C.1. Proof of Proposition 3.2

We prove the proposition by introducing the following theorem first.

**Theorem C.1** (Theorem 1 of (Micali & Zhu, 2016)). *Given a graph $\mathcal{G} = (\mathcal{V}, \mathcal{E})$, one can reconstruct $\mathcal{B}(v, k) = (\mathcal{V}', \mathcal{E}')$, where $n = |\mathcal{V}'|, m = |\mathcal{E}'|$, the ego-graph induced by node $v \in \mathcal{V}$ with $k$ radius, via an anonymous walk distribution $\mathcal{D}_l$, where $l = O(m)$ starting at node $v$.*

As stated in the theorem, let $\mathcal{D}_l$ denote the distribution of anonymous walks sampled from a node $v$. This distribution is sufficient to represent the complete $k$-hop ego-graph induced from $v$, $\mathcal{B}(v, k) = (\mathcal{V}', \mathcal{E}')$, where $l = O(|\mathcal{E}'|)$. In other words, if a set of patterns $\{\phi_1, \phi_2, \dots\}$ can approximate $\mathcal{D}_l$, this set can be used to reconstruct the corresponding ego-graph. Thus, for each node, a set of $l$-length patterns is sufficient to reconstruct its respective $k$-hop ego-graph.

Next, we consider the scenario where a task requires information from the $k$-hop ego-graph $\mathcal{B}(v, k)$. Given that the ego-graph distribution of each node can be reconstructed via anonymous walks, it becomes feasible to compare these distributions to assess whether node representations are distinguishable. In general, the $k$-hop ego-graph distributions of two nodes $u$ and $v$ are distinct unless they represent the same structural phenomena. Consequently, the corresponding pattern sets $\{\phi_1^u, \phi_2^u, \dots\}$ and $\{\phi_1^v, \phi_2^v, \dots\}$ are also distinct, ensuring that each node retains a unique and distinguishable topological representation.

### C.2. Proof of Proposition 3.3

As discussed, any graph pattern can be represented as a combination of a semantic path, which captures semantic information (i.e., the specific nodes forming the pattern), and an anonymous path, which encodes topological structure (i.e., the overall pattern structure). To effectively extract both types of information, these two paths can be encoded separately, preserving semantic meaning and structural insight independently.

To ensure lossless compression, we employ bijective mappings to project these paths, guaranteeing that distinct paths maintain unique representations. Given a semantic path $w$ and its corresponding anonymous path $\phi$, we introduce two bijective projections: $\rho_s : w \to \boldsymbol{p}_s$ for semantic encoding and $\rho_a : \phi \to \boldsymbol{p}_a$ for structural encoding. Consequently, to comprehensively encode a given graph pattern, both the semantic and anonymous paths must be jointly represented.

### C.3. Proof of Theorem 3.4

We outline the proof by first (1) establishing the expressiveness of a simplified variant of GPM and (2) extending this result to the general case.

The learning process of GPM consists of three key steps: (1) Extracting $n$ patterns using $l$-length random walks, where each pattern is uniquely defined by its semantic path $w$ and anonymous path $\phi$. (2) Encoding the semantic and anonymous paths separately using neural networks and combining their representations. (3) Passing the encoded graph patterns through a transformer for final predictions.

To analyze expressiveness, we consider a simplified variant of GPM with the following modifications: (1) Setting $n = 1$. (2) Replacing neural networks with universal and injective mappings $\rho_s$ and $\rho_a$. (3) Using a mean aggregator over encoded patterns instead of a transformer.

Under this setting, the model is essentially trained on a single $l$-length path $w$. Additionally, we impose the following mild assumptions:

**Assumption C.2.** The graphs are connected, unweighted, and undirected.

**Assumption C.3.** The walk length $l$ is sufficiently large.

Given these assumptions, the simplified GPM can distinguish non-isomorphic graphs, as stated in the following theorem.

**Theorem C.4** (Theorem 4 of Wang & Cho (2024)). *Up to the Reconstruction Conjecture, encoding the $l$-length random walks (combining semantic path and anonymous path) produces distinct embeddings for non-isomorphic graphs.*

The key insight is that for any two non-isomorphic graphs, the distributions of infinite-length random walks over these graphs are distinct, regardless of the starting points. Moreover, universal and injective mappings ensure that each unique random walk is projected into a unique point in the embedding space. Wang & Cho (2024) establishes this theorem by (1) proving it for the simplest case where the graph size is 3, and (2) using induction to extend the proof to graphs of size $n - 1$ and $n$. Based on this theorem, it is straightforward to demonstrate that the simplified GPM can distinguish any non-isomorphic graphs, provided the reconstruction conjecture holds. Next, we generalize the simplified case to the proposed GPM framework.

(1) The simplified case assumes $n = 1$ with a sufficiently large walk length $l$, whereas GPM operates with $n > 1$. Since random walks can start from any node in the graph, an $l$-length random walk can be split into $k$ segments ($k$ is large enough), each of length $l/k$. Each sub-walk can be encoded individually and later combined to approximately form the final embedding. Note that We did not mean to suggest that long walks can be fully reconstructed from shorter ones, especially

since anonymous walks cannot preserve node identity across segments. Rather, we intended to describe an approximate strategy, where long walks are segmented into shorter sub-walks, each encoded independently. This design allows the transformer to aggregate distributed long-range information across these sub-patterns. (2) GPM employs neural networks as encoders, which are inherently universal and injective, satisfying the requirements of the mappings $\rho_s$ and $\rho_a$ used in the simplified case. (3) Finally, while the simplified case uses a mean aggregator over encoded patterns, GPM adopts a transformer architecture for aggregation. The self-attention mechanism in transformers generalizes the mean aggregator, which can be seen as a special case of the transformer.

By these generalizations, we establish that GPM can distinguish any connected, unweighted, and undirected non-isomorphic graphs, given a sufficient number of graph patterns.

### C.4. Proof of Theorem 3.5

The proof follows the same structure as Theorem 3.4: (1) Defining a simplified variant of GPM, (2) Proving the theorem on this simplified model, and (3) Extending the results to the general case.

To ensure the proof is self-contained, we reintroduce the design of the simplified variant and its extension to the full model.

The simplified variant of GPM includes the following modifications: (1) Replacing neural networks with universal and injective mappings $\rho_s$ and $\rho_a$. (2) Using a mean aggregator over encoded patterns instead of a transformer.

Under the same assumptions as in Theorem 3.4, namely: (1) The graphs are connected, unweighted, and undirected, and (2) The number of sampled patterns is sufficiently large, we apply the following corollary to directly prove Theorem 3.5 for the simplified case.

**Theorem C.5** (Corollary 4.1 of Wang & Cho (2024)). *Up to the Reconstruction Conjecture, two graphs $\mathcal{G}_1, \mathcal{G}_2$ labeled as non-isomorphic by the $k$-WL test is the necessary, but not sufficient condition that encoding the $k$-length random walks sampled from these two graphs produces the same embedding.*

In other words, if the $k$-WL test distinguishes two graphs, then the simplified GPM variant can also distinguish them. However, the converse does not necessarily hold—if the simplified GPM distinguishes two graphs, the $k$-WL test may fail to do so.

To generalize from the simplified case to the full GPM model, we follow the same strategy as in the proof of Theorem 3.4: (1) Replacing the universal and injective mappings with neural network encoders, and (2) Substituting the mean aggregator with a transformer architecture.

## D. Complexity

We analyze the time complexity of three key components: pattern sampling, pattern encoding, and transformer encoding. For a single instance, $k$ random walks of length $l$ are sampled, resulting in a sampling complexity of $O(k \cdot l)$, where $l^2 \approx k$ empirically. Pattern encoding involves three alternative encoders: the mean encoder with complexity $2 \times O(k)$, the GRU encoder with complexity $2 \times O(k \cdot l)$, and the transformer encoder with complexity $2 \times O(k \cdot l^2)$ (note both semantic and anonymous paths should be encoded). The tranformer encoder over encoded graph patterns introduces an additional complexity of $O(k^2)$. Thus, the maximum total complexity per instance is

$$O(k \cdot l) + 2 \times O(k \cdot l^2) + O(k^2) \approx O(k^2).$$

We compare this complexity to that of existing graph transformers for both node-level and graph-level tasks. For node classification, GPM's time complexity for encoding all nodes in a graph is $O(n \cdot k^2)$, where $n$ is the number of nodes, and $k^2 \ll n$. In contrast, existing graph transformers incur $O(n^2)$ complexity, which scales quadratically with the number of nodes. For graph classification, GPM's complexity for encoding all graphs is $O(m \cdot k^2)$, where $m$ is the number of graphs. In comparison, existing graph transformers require $O(m \cdot n^2)$, where $n$ is the average number of nodes per graph.

# E. How Does GPM Surpass Message Passing?

## E.1. Empirical Effectiveness

Beyond theoretical analysis, we provide an empirical evaluation on three benchmark datasets specifically designed to challenge graph isomorphism tests. The CSL dataset (Murphy et al., 2019) comprises 150 4-regular graphs that are indistinguishable using the 1-WL test. The EXP dataset (Abboud et al., 2021) includes 600 pairs of non-isomorphic graphs that cannot be distinguished by either the 1-WL or 2-WL tests. Lastly, the SR25 dataset (Balcilar et al., 2021) contains 15 strongly regular graphs with 25 nodes each, which remain indistinguishable even under the 3-WL test. The experimental results, summarized in Table 8, demonstrate that GPM successfully distinguishes all graphs across these datasets, empirically surpassing the 3-WL test. In contrast, many existing models fail on these tasks.

*Table 8.* Empirical expressiveness analysis.

|  | CSL | EXP | SR25 |
| --- | --- | --- | --- |
| GCN (Kipf & Welling, 2017) | 10 | 51.9 | 6.7 |
| GatedGCN (Bresson & Laurent, 2017) | 10 | 51.7 | 6.7 |
| GraphTrans (Dwivedi & Bresson, 2020) | 10 | 52.4 | 6.7 |
| 3-GNN (Morris et al., 2019) | 95.7 | 99.7 | 6.7 |
| GIN-AK+ (Zhao et al., 2022) | - | 100 | 6.7 |
| GraphViT (He et al., 2023) | 100 | 100 | 100 |
| ESC-GNN (Yan et al., 2024) | 100 | 100 | 100 |
| GPM | 100 | 100 | 100 |

## E.2. Tackling Over-Squashing

Another limitation of message passing is their focus on localized information, which prevents them from effectively capturing long-range dependencies within graphs. In contrast, GPM demonstrates superior capability in modeling long-range interactions. Following He et al. (2023), we evaluate this capability using the TREENEIGHBORSMATCHING benchmark introduced by Alon & Yahav (2021). This dataset comprises binary trees, with the objective being to classify the root node based on its degree. The degree information is maintained within the leaf nodes, and successful classification of the root node requires the model to capture $r$-radius information, where $r$ is the depth of the tree.

As shown in Figure 7, we report the training accuracy following Alon & Yahav (2021) to evaluate the model's fitting ability. Notably, message passing GNNs and RUM (Wang & Cho, 2024) fail to perfectly fit the data, exhibiting over-squashing effects as early as $r = 4$. In other words, these models fail to distinguish between different training examples, even when these examples are observed multiple times. In contrast, GPM achieves perfect data fitting across all problem radiuses. This is attributed to its ability to directly learn long-range patterns sampled via random walks.

To provide a clearer understanding, we present a brief example illustrating how GPM mitigates over-squashing in such scenarios. Recall that the correlation between node labels and degrees is encoded within the leaf nodes, where the features of the leaf nodes are a combination of their degrees and labels (for simplicity). Even the simplest variant of GPM (without anonymous path encoding or node positional embeddings) has the potential to solve this problem. Specifically, given a tree

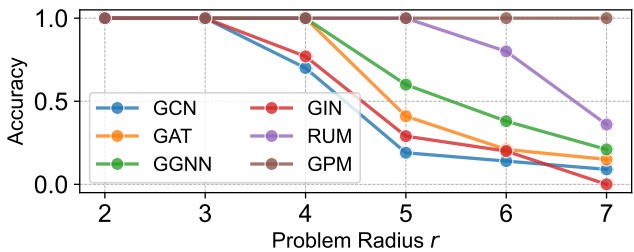

*Figure 7.* Over-squashing analysis.

with radius $r$, we randomly sample $k$ random walks of length $r$ and encode these walks using transformer. In the optimal (and empirically observed) case, the encoded pattern embeddings accurately capture the features of the leaf nodes, i.e., the combination of node features and labels. At this stage, the model effectively reduces the task to a basic matching problem: given multiple sets of (node degree, label) pairs, it predicts the label of the root node based on its degree. This matching task is straightforwardly handled by neural networks, such as the transformer architecture employed in our work.

## F. Additional Experimental Results and Discussions

### F.1. Additional Discussion on Distribution Training

We use GraphSAGE as the GNN baseline, fixing the batch size to 256 and the number of training epochs to 10. Distributed GNN training follows the graph partitioning without replication strategy (Cai et al., 2021), where the graph is divided into non-overlapping partitions using the METIS library (Karypis, 1997). Each GPU processes a single partition, and peer-to-peer communication is used to exchange learned node embeddings among GPUs.

Due to resource limitations, our experiments are conducted on a single machine equipped with four Nvidia A40 GPUs, which does not reflect acceleration performance in multi-machine distributed settings. However, Shoeybi et al. (2019) demonstrated that the scalability of transformer architectures improves consistently with an increasing number of computational devices. In contrast, Cai et al. (2021) showed that the graph partitioning without replication strategy introduces substantial communication overhead in GNNs as the number of GPUs increases. Specifically, Cai et al. (2021) (Figure 2) reports that when scaling to 16 GPUs, the training becomes slower than with just 2 GPUs due to communication overhead dominating time consumption.

These findings highlight the advantages of GPM, which using tranformer as backbone, over message passing GNNs for distributed training, particularly in scenarios with a large number of devices.

### F.2. Model Component Ablation

To verify the contribution of key components in GPM, we conduct several ablation studies, as summarized in Table 9.

**Impact of Positional Embeddings and Anonymous Paths.** We investigate the role of positional embeddings and anonymous paths in GPM. While both provide topological insights, positional embeddings primarily capture node-level structural information (i.e., the relative position of a node within the graph), whereas anonymous paths encode structural characteristics of patterns (i.e., the identity and structure of the pattern itself). As shown in Table 9, incorporating positional embeddings and anonymous paths improves model performance. However, the effect of positional embeddings is relatively

*Table 9.* Full results on model component ablation.

| | Task | PRODUCTS Node | COMPUTER Node | ARXIV Node | WIKICS Node | CORAFULL Node | DEEZER Node | BLOG Node |
|---|---|---|---|---|---|---|---|---|
| Comp. | GPM | **82.62 ± 0.39** | **92.28 ± 0.39** | **72.89 ± 0.68** | **80.19 ± 0.41** | **71.23 ± 0.51** | 67.26 ± 0.22 | **96.71 ± 0.59** |
| | w/o PE | 82.47 ± 0.21 | 91.89 ± 0.64 | 72.59 ± 0.23 | 80.02 ± 0.43 | 70.71 ± 0.50 | **67.82 ± 0.39** | 96.38 ± 0.60 |
| | w/o AP | 81.99 ± 0.68 | 91.83 ± 0.55 | 72.47 ± 0.40 | 79.56 ± 0.43 | 70.42 ± 0.36 | 66.90 ± 0.17 | 92.10 ± 1.72 |
| | w/o PE & AP | 80.74 ± 0.65 | 91.14 ± 0.30 | 71.40 ± 0.76 | 79.89 ± 0.41 | 70.69 ± 0.48 | 66.24 ± 0.62 | 90.74 ± 1.06 |
| SP Enc | Mean | 80.42 ± 0.48 | 91.11 ± 0.49 | 72.08 ± 0.66 | 79.82 ± 0.42 | **71.25 ± 0.54** | 64.62 ± 0.21 | 90.58 ± 0.48 |
| | GRU | 80.91 ± 0.33 | 90.05 ± 0.71 | 72.23 ± 0.41 | 79.54 ± 0.41 | 70.92 ± 0.56 | 65.88 ± 0.11 | 91.88 ± 2.93 |
| | Transformer | **82.62 ± 0.39** | **92.28 ± 0.39** | **72.89 ± 0.68** | **80.19 ± 0.41** | 71.23 ± 0.51 | 67.26 ± 0.22 | **96.71 ± 0.59** |

| | Task | FLICKR Node | FLICKR-S Node | OGBL-COLLAB Link | IMDB-B Graph | COLLAB Graph | REDDIT-M5K Graph | REDDIT-M12K Graph |
|---|---|---|---|---|---|---|---|---|
| Comp. | GPM | 52.22 ± 0.19 | **89.41 ± 0.47** | **49.70 ± 0.59** | **82.67 ± 0.47** | **80.70 ± 0.74** | **51.87 ± 1.04** | **43.07 ± 0.29** |
| | w/o PE | **52.26 ± 0.20** | 88.45 ± 0.80 | 49.59 ± 0.65 | 79.00 ± 1.63 | 78.00 ± 1.07 | 50.88 ± 0.78 | 41.73 ± 0.83 |
| | w/o AP | 51.52 ± 0.12 | 88.14 ± 0.53 | 48.43 ± 0.97 | 80.48 ± 2.18 | 78.33 ± 0.24 | 49.08 ± 1.51 | 39.73 ± 1.09 |
| | w/o PE & AP | 51.39 ± 0.01 | 87.70 ± 0.40 | 48.24 ± 0.97 | 81.33 ± 3.30 | 75.40 ± 0.99 | 48.73 ± 0.25 | 39.50 ± 0.46 |
| SP Enc | Mean | 51.43 ± 0.06 | 89.86 ± 0.26 | 47.83 ± 1.00 | **83.33 ± 0.94** | 76.27 ± 0.47 | 47.30 ± 2.39 | 40.78 ± 0.31 |
| | GRU | 51.12 ± 0.06 | **90.13 ± 0.31** | 48.12 ± 0.55 | 82.00 ± 2.83 | 74.00 ± 1.57 | 47.12 ± 1.44 | 42.44 ± 0.83 |
| | Transformer | **52.22 ± 0.19** | 89.41 ± 0.47 | **49.70 ± 0.59** | 82.67 ± 0.47 | **80.70 ± 0.74** | **51.87 ± 1.04** | **43.07 ± 0.29** |

marginal, while the impact of anonymous paths is more pronounced. This discrepancy suggests that GPM, which learns from graph patterns rather than individual nodes, benefits more from understanding the structural composition of patterns than from knowing the specific locations of nodes within them. Notably, when both components are removed, model performance deteriorates significantly, highlighting the necessity of topological information for effective learning.

**Impact of Semantic Path Encoders.** We analyze the effect of different semantic path encoders, including Mean, GRU, and Transformer. The Transformer encoder achieves the best performance on 11 out of 14 datasets, owing to its inherent ability to model both localized and long-range dependencies. Comparing Mean and GRU, we observe that GRU outperforms Mean on heterophilous graphs (e.g., Deezer, Blog, Flickr-S), likely due to its recurrent structure, which better captures long-range dependencies in the data.

### F.3. Additional Discussion on Learning Strategies

**Multi-Scale Learning.** Table 10 presents the model performance with and without multi-scale training. The results demonstrate that multi-scale training generally enhances performance across various tasks, with the exception of heterophilous graphs (Blog, Flickr, Flickr-S). This suggests that sampling graph patterns of varying sizes effectively captures different levels of information dependencies. However, in heterophilous graphs, which inherently favor long-range dependencies, sampling smaller patterns may overemphasize localized structures, leading to performance degradation.

*Table 10.* Impact of multi-scaling training.

| Task | PRODUCTS
Node | COMPUTER
Node | ARXIV
Node | WIKICS
Node | CORAFULL
Node | DEEZER
Node | BLOG
Node |
|---|---|---|---|---|---|---|---|
| GPM | **82.62 ± 0.39** | **92.28 ± 0.39** | **72.89 ± 0.68** | **80.19 ± 0.41** | **71.23 ± 0.51** | **67.26 ± 0.22** | 96.71 ± 0.59 |
| w/o multi-scale | 82.44 ± 0.20 | 91.00 ± 0.42 | 71.79 ± 0.40 | 80.05 ± 0.32 | 62.55 ± 0.52 | 64.56 ± 0.16 | **97.02 ± 0.47** |

| Task | FLICKR
Node | FLICKR-S
Node | OGBL-COLLAB
Link | IMDB-B
Graph | COLLAB
Graph | REDDIT-M5K
Graph | REDDIT-M12K
Graph |
|---|---|---|---|---|---|---|---|
| GPM | 52.22 ± 0.19 | 89.41 ± 0.47 | **49.70 ± 0.59** | **82.67 ± 0.47** | **80.70 ± 0.74** | **51.87 ± 1.04** | **43.07 ± 0.29** |
| w/o multi-scale | **52.26 ± 0.14** | **90.68 ± 0.79** | 48.30 ± 0.39 | 82.33 ± 1.89 | 77.73 ± 0.90 | 47.87 ± 0.98 | 41.46 ± 0.40 |

**Test-Time Augmentation.** Table 11 presents the impact of test-time augmentation in GPM. We evaluate three variants: (1) *# Train=16, # Infer=128*, where 16 patterns are used during training and 128 during inference (the default setting); (2) *# Train=16, # Infer=16*, which uses 16 patterns for both training and inference, serving as the lower bound of GPM; and (3) *# Train=128, # Infer=128*, where 128 patterns are used in both training and inference, representing the upper bound of GPM.

Comparing *# Train=16, # Infer=16* with *# Train=128, # Infer=128*, we observe that increasing the number of patterns significantly improves performance but also incurs substantial computational overhead (Table 12). To balance high performance with computational efficiency in training, we adopt *# Train=16, # Infer=128*, where fewer patterns are used during training while maintaining a larger number during inference. Interestingly, in some cases, this variant even outperforms *# Train=128, # Infer=128*, possibly due to the reduced risk of overfitting of fewer patterns in training.

*Table 11.* Impact of test-time augmentation.

| Task | PRODUCTS
Node | COMPUTER
Node | ARXIV
Node | WIKICS
Node | CORAFULL
Node | DEEZER
Node | BLOG
Node |
|---|---|---|---|---|---|---|---|
| # Train=16 # Infer=128 | **82.62 ± 0.39** | 92.28 ± 0.39 | **72.89 ± 0.68** | 80.19 ± 0.41 | 71.23 ± 0.51 | 67.26 ± 0.22 | **96.71 ± 0.59** |
| # Train=16 # Infer=16 | 80.89 ± 0.23 | 90.10 ± 0.48 | 69.90 ± 0.00 | 78.45 ± 0.55 | 62.57 ± 0.35 | 64.54 ± 0.27 | 86.53 ± 0.57 |
| # Train=128 # Infer=128 | 82.39 ± 0.50 | **92.54 ± 0.74** | 72.42 ± 0.00 | **80.83 ± 0.50** | **71.24 ± 0.06** | **67.71 ± 0.09** | 96.48 ± 0.24 |

| Task | FLICKR
Node | FLICKR-S
Node | OGBL-COLLAB
Link | IMDB-B
Graph | COLLAB
Graph | REDDIT-M5K
Graph | REDDIT-M12K
Graph |
|---|---|---|---|---|---|---|---|
| # Train=16 # Infer=128 | 52.22 ± 0.19 | **89.41 ± 0.47** | 49.70 ± 0.59 | 82.67 ± 0.47 | 80.70 ± 0.74 | **51.87 ± 1.04** | 43.07 ± 0.29 |
| # Train=16 # Infer=16 | 50.07 ± 0.08 | 81.15 ± 1.42 | 47.30 ± 0.69 | 81.33 ± 1.25 | 76.27 ± 0.38 | 44.53 ± 0.81 | 39.12 ± 0.62 |
| # Train=128 # Infer=128 | **52.85 ± 0.28** | 89.28 ± 0.42 | **49.92 ± 0.53** | **82.87 ± 3.05** | **80.93 ± 0.81** | 51.27 ± 0.93 | **43.27 ± 0.04** |

*Table 12.* Training time (second per epoch) and acceleration of using less patterns.

|  | PRODUCTS | COMPUTER | ARXIV | WIKICS | FLICKR |
|---|---|---|---|---|---|
| # Train = 128 | 682.07s | 19.15s | 179.78s | 0.92s | 83.36s |
| # Train = 16 | 44.58s | 1.22s | 12.84s | 0.08s | 5.54s |
| Acceleration | ×15.30 | ×15.70 | ×14.00 | ×11.50 | ×15.05 |

## G. GPM Automatically Learns Data Dependencies

Graph datasets often exhibit a mixture of localized and long-range dependencies. While social networks are commonly assessed to preserve localized dependencies (Granovetter, 1973; Liben-Nowell & Kleinberg, 2003), certain tasks demand an understanding of long-range dependencies. For instance, detecting rumor spreaders involves tracing information flow across the network (Bian et al., 2020), as rumors can propagate through multiple intermediaries, with the origin potentially disconnected from many affected nodes.

Leveraging the transformer architecture, GPM can automatically identify dominant patterns (no matter localized or long-range) relevant to downstream tasks. This capability allows the model to learn the underlying data dependencies within the graph. Such a property not only reduces the need for extensive hyperparameter tuning, where other models might require different hyperparameter settings to capture varying dependencies, but also enhances overall performance.

We conduct experiments on the PRODUCTS dataset. By leveraging biased random walks for pattern sampling, we control the sampling bias to generate mixed ($p = 1, q = 1$), localized ($p = 0.1, q = 10$), or long-range ($p = 10, q = 0.1$) patterns. Figure 5 presents the training loss and testing accuracy under these settings. The results show that models using unbiased random walks ($p = 1, q = 1$), which uniformly sample patterns with both localized and long-range dependencies, achieve the best performance. This validates that the model can autonomously determine the importance of different patterns. In contrast, when data dependencies are pre-determined ($p = 0.1, q = 10$ or $p = 10, q = 0.1$), the model may fail to capture the most relevant patterns for each instance, leading to degraded performance.

An intriguing and counterintuitive observation is that models utilizing long-range patterns outperform those relying on localized patterns, even on graphs with high homophily ratios. This phenomenon, consistently observed across other datasets, can be attributed to the large degree of duplication in sampled localized patterns, which arises from the long-tail degree distribution of nodes. Such duplications hinder the model to identify truly dominant patterns, impairing performance. In contrast, long-range patterns, while containing more noise, are less redundant and have a higher probability of including the dominant patterns. However, the noise inherent in long-range sampling still prevents optimal performance. Based on these observations, we hypothesize that unbiased sampling achieves a balance between redundancy and noise, enabling the model to learn more effectively.

## H. Model Interpretation on COMPUTERS

See Figure 8.

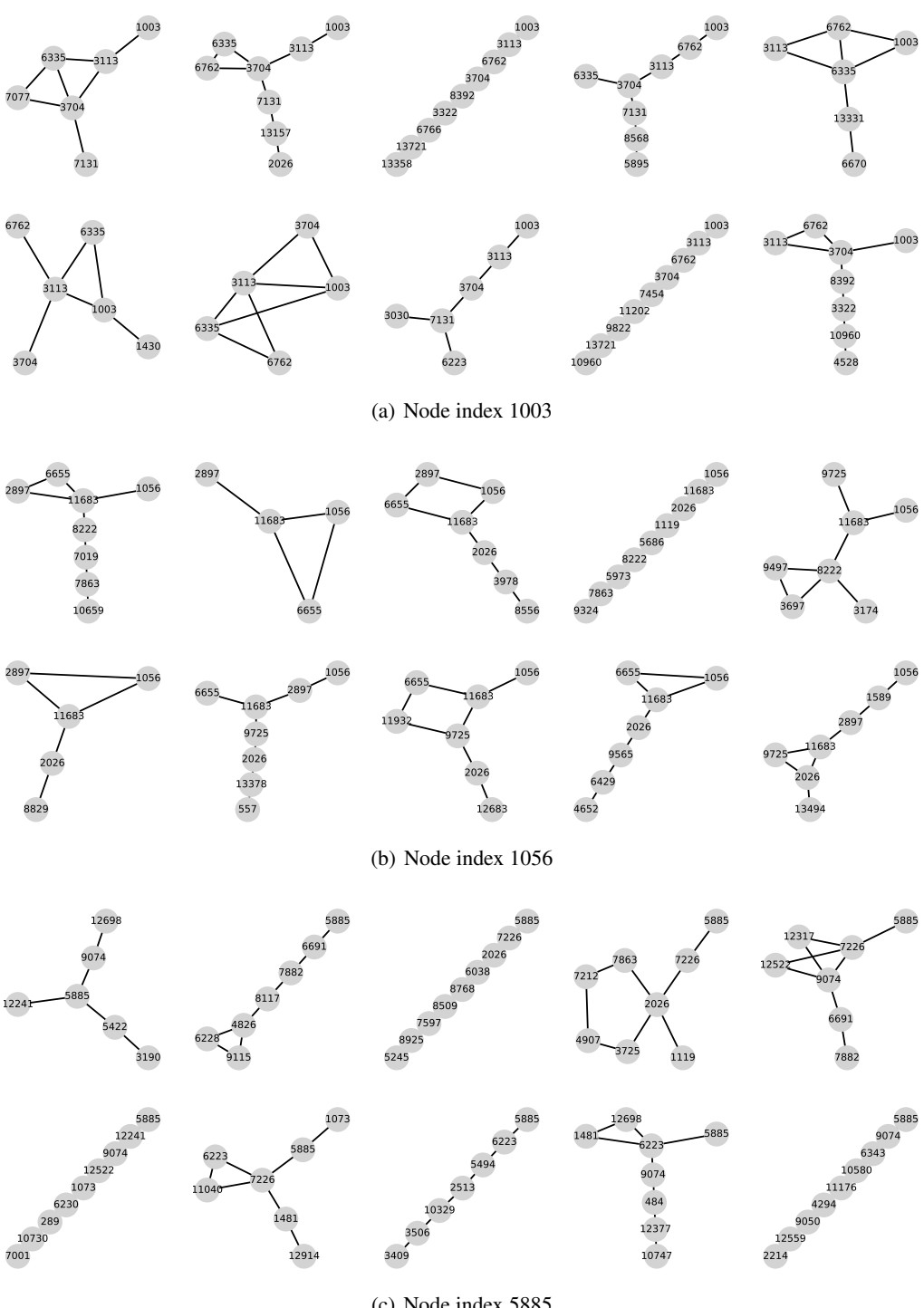

Figure 8. The top-10 important patterns associated to the certain nodes.

