# OpenReview forum: "Beyond Message Passing: Neural Graph Pattern Machine"
_ICML.cc/2025/Conference — ICML 2025 poster_

### Official Review · Reviewer_ykrV · 2025-02-21

**Overall Recommendation:** 3

**Summary:**

## Update after rebuttal
The authors have solved most of my concerns. I decide to maintain my score.

Graph neural networks struggle to capture essential substructures, such as triangles in social networks or benzene rings in molecular graphs, due to their reliance on message passing. To address this limitation, the Neural Graph Pattern Machine (GPM) is introduced as a framework that directly learns from graph patterns, efficiently extracting and encoding substructures while identifying those most relevant to downstream tasks. Theoretically, GPM surpasses message passing in expressivity and long-range information modeling.  The authours have conducted extensive experiments across multiple graph learning tasks to demonstrate its superiority over state-of-the-art baselines.

**Claims And Evidence:**

Yes, the claims made in the submission are supported by clear and convincing evidence.

**Essential References Not Discussed:**

No

**Experimental Designs Or Analyses:**

See the weakness.

**Methods And Evaluation Criteria:**

Yes

**Other Comments Or Suggestions:**

No.

**Other Strengths And Weaknesses:**

Strengths:
Refer to the summary.

Weaknesses:
1. The fundamental framework of GPM appears similar to [1]. It is recommended that the authors provide a detailed comparison to clarify the distinctions between their approach and this prior work.

2. Regarding the Pattern Encoder, numerous studies have explored the integration of Random Walk with GNNs/RNNs, including DeepWalk [2], GraphRNA [3], and RAW-GNN [4]. The authors should explicitly highlight the key differences between GPM and these existing methods.

3. In Theorem 3.4, GPM is shown to distinguish non-isomorphic graphs given a sufficient number of graph patterns. However, how is this number determined? Are there theoretical bounds on the required number of patterns? Addressing this would be crucial, as it represents a trade-off between computational complexity and learning sufficiency.

[1] Wang Y, Cho K. Non-convolutional graph neural networks[J]. arXiv preprint arXiv:2408.00165, 2024.
[2] Perozzi B, Al-Rfou R, Skiena S. Deepwalk: Online learning of social representations[C]//Proceedings of the 20th ACM SIGKDD international conference on Knowledge discovery and data mining. 2014: 701-710.
[3] Huang X, Song Q, Li Y, et al. Graph recurrent networks with attributed random walks[C]//Proceedings of the 25th ACM SIGKDD International Conference on Knowledge Discovery & Data Mining. 2019: 732-740.
[4] Jin D, Wang R, Ge M, et al. Raw-gnn: Random walk aggregation based graph neural network[J]. arXiv preprint arXiv:2206.13953, 2022.

**Questions For Authors:**

See the weaknesses.

**Relation To Broader Scientific Literature:**

See the weakness.

**Theoretical Claims:**

Yes, see the weakness.

---

> ### Author Rebuttal · Authors · 2025-03-30
>
> Thank you for the thoughtful review. We appreciate your constructive suggestions regarding related work and theoretical clarification, and we address each of these points in detail below.
>
> > A detailed comparison to RUM.
> >
>
> **Motivation**: RUM is designed to jointly address expressiveness, over-smoothing, and over-squashing within a message-passing-motivated framework (see below). In contrast, GPM is motivated by **bypassing message passing** and learning directly from **substructure patterns**.
>
> **Graph Inductive Bias**: While both RUM and GPM use **semantic paths** and **anonymous paths** to represent individual walks, their modeling approaches diverge significantly. RUM utilizes **RNNs (GRUs)** to encode these paths, following a philosophy aligned with message passing—where localized information dominates importance. GPM, however, leverages **Transformers** to encode semantic paths, allowing the model to **learn inductive biases directly from data**, particularly important in tasks requiring **long-range dependency modeling**.
>
> **Pattern Aggregator**: GPM employs a **Transformer-based aggregator** to combine learned graph pattern representations, whereas RUM uses a **mean aggregator**. This design choice brings several advantages: **(1) Scalability**: GPM scales effectively to large graphs and large model sizes (as demonstrated in Tables 1–3 and Figure 4). **(2)** **Effectiveness**: GPM samples 128 graph patterns per instance during inference (vs. 4 in RUM), and the Transformer allows the model to **attend to the most informative patterns**, yielding superior performance across benchmarks (Tables 1–4). **(3) Interpretability**: The attention mechanism in GPM’s Transformer provides **natural interpretability**, allowing us to identify key substructure patterns (e.g., via attention weights in Figure 5), a feature absent in RUM.
>
> **Empirical Comparison:** The distinctions above enables our GPM consistently outperforms RUM empirically (see Table 1,2,3,4). For example, in large-scale ogbn-product dataset (about 2.5 million nodes), GPM achieves 82.62% accuracy comparing to 78.68% of RUM.
>
> > Explicitly highlight the key differences between GPM and DeepWalk, GraphRNA, RAW-GNN.
> >
>
> We thank the reviewer for this request and provide the following comparison. **DeepWalk and GraphRNA** follow the principle of representing an entire graph as a collection of random walks that capture **co-occurrence relationships among nodes**. Their differences mainly lie in how walks are generated and encoded.
>
> **GPM and RAW-GNN**, in contrast, adopt the philosophy that **random walks represent individual substructures** of graph instances (nodes, edges, or entire graphs). The key distinctions between GPM and RAW-GNN are (1) **Sampling Strategy**: GPM uses unbiased random walks, reducing inductive bias. RAW-GNN employs biased sampling, which may impose structural priors that limit generalizability. (2) **Structural Representation**: GPM explicitly **decouples topology and semantics** by representing each walk as semantic and anonymous paths. RAW-GNN does not separate structural and feature encoding in this way. (3) **Pattern Aggregation**: GPM uses a **single** Transformer to aggregate all patterns. RAW-GNN uses **separate** aggregators for low- and high-order patterns, which may prevent modeling interactions across different structural levels.
>
> **Empirical Comparison:** Based on the above limitations, GPM consistently outperforms these methods empirically. We show the comparison results on node classification in the following.
>
> | **Dataset** | **DeepWalk** | **GraphRNA** | **RAW-GNN** | **GPM (ours)** |
> | --- | --- | --- | --- | --- |
> | **Computers** | 88.56 | 91.06 | 92.03 | **92.28** |
> | **WikiCS** | 76.34 | 77.65 | 79.01 | **80.19** |
> | **Flickr** | 49.70 | 49.78 | 49.58 | **52.22** |
>
> > Theoretical clarification
> >
>
> Theorem 3.4 demonstrates the **existence** of a sufficient number of graph patterns such that GPM can distinguish any pair of non-isomorphic graphs under its pattern-based encoding. However, this is a **non-constructive** result—it does not provide an explicit bound on the required number of patterns. Establishing such a bound is challenging due to the **combinatorial nature** of graph structures and the fact that the number may depend on: (1) Graph size and topological complexity, (2) The specific classes of non-isomorphic graphs (e.g., regular vs. irregular), (3) The expressiveness of the semantic and anonymous path encoders.
>
> Despite the theoretical uncertainty, our empirical findings show that **sampling 128 random walks per graph** is often sufficient to achieve **perfect accuracy** in distinguishing non-isomorphic graphs (see Table 8).
>
> We agree that deriving a **tight theoretical bound** on the number of necessary patterns is an important direction for future work, and we appreciate the reviewer for highlighting this.

---

> > ### Comment · Reviewer_ykrV · 2025-04-03
> >
> > Thanks for the response from authors. I decide to maintain my score.

---

> > > ### Author Response · Authors · 2025-04-06
> > >
> > > Thank you for your response. Your feedback is genuinely appreciated and helps us continue to improve our work.

---

### Official Review · Reviewer_Ynb7 · 2025-03-09

**Overall Recommendation:** 3

**Summary:**

The paper introduces the Neural Graph Pattern Machine (GPM), a framework designed to enhance the expressiveness of graph learning models by directly learning from graph patterns. Traditional Graph Neural Networks (GNNs) rely on message passing to aggregate information from local neighborhoods, which can limit their ability to identify fundamental substructures, such as triangles. GPM addresses this limitation by efficiently extracting and encoding substructures, identifying the most relevant ones for downstream tasks. Empirical evaluations across various tasks, including node classification, link prediction, graph classification, and regression, demonstrate GPM's superiority over state-of-the-art baselines. The paper also highlights GPM's robustness, scalability, and interpretability, offering a comprehensive analysis of its performance. Overall, the paper is well-organized and clearly presented.

**Claims And Evidence:**

Yes

**Essential References Not Discussed:**

no

**Experimental Designs Or Analyses:**

yes
In Section 4.3, the authors highlight that traditional message-passing Graph Neural Networks (GNNs), such as the Graph Attention Network (GAT), encounter scalability challenges due to the oversmoothing effect. This phenomenon occurs when node representations become indistinguishable as the number of GNN layers increases, ultimately diminishing model performance. To address this issue, the proposed Neural Graph Pattern Machine (GPM) takes a novel approach by learning directly from graph patterns instead of relying solely on message passing. By efficiently extracting and encoding substructures, GPM strives to preserve distinctive node representations. Therefore, it should have the potential to overcome the oversmoothing problem, (if my understanding is correct). However, further experiments are needed to substantiate this claim.

**Methods And Evaluation Criteria:**

yes

**Other Comments Or Suggestions:**

1. Strongly recommend the author publicly disclose the code to facilitate reproducibility.

2. Suggest adding experiments on heterogeneous graphs. While the author mentions this in the limitations section, evaluating GPM's classification potential on heterogeneous graphs would be valuable.

3. Recommend increasing experimental comparisons to address oversmoothing. For instance, increasing the number of network layers to evaluate model performance would be beneficial.

4. How does the model perform on large datasets, such as OGB? Including an assessment of GPM's performance on big datasets would strengthen the paper.

**Other Strengths And Weaknesses:**

The main shortcomings of this paper lie in its reliance on existing technology. However, the author has provided valuable insights into the rationale behind using this technology, supported by corresponding evidence, and conducted thorough experiments. Including experiments specifically aimed at overcoming the oversmoothing issue would further enhance the paper's contributions.

**Questions For Authors:**

How does the model perform on large datasets, such as OGB?
How does the model perform on the heterogeneous graph dataset?

**Relation To Broader Scientific Literature:**

The key contributions of this paper relate to the broader scientific literature as follows:

New Approach: The framework, GPM, overcomes the limitations of traditional GNNs like restricted expressiveness and oversquashing, echoing ongoing research efforts in addressing these issues.

Scalability: GPM scales better for complex tasks involving large models and graphs, contributing to the field's discussions on handling vast datasets.

Interpretability: GPM provides a way to interpret models, which is crucial for applications like social network analysis and drug discovery. This aligns with the growing focus on the interpretability of GNNs in the literature.

**Theoretical Claims:**

yes

---

> ### Author Rebuttal · Authors · 2025-03-30
>
> Thank you for the thoughtful and encouraging review. We’re glad the reviewer appreciated our contributions in expressiveness, scalability, and interpretability. We also appreciate the constructive suggestions regarding additional experiments, and we address each point in detail below.
>
> > GPM has the potential to overcome over-smoothing. Need more experiments.
> >
>
> Absolutely — we also believe that GPM naturally mitigates the over-smoothing problem. In traditional message-passing GNNs, deeper networks tend to aggregate node features from increasingly distant neighbors, often leading to over-smoothing. In GPM, the analogous hyperparameter is the **random walk length**, which controls the receptive field of each pattern.
>
> To investigate this, we conducted experiments on **WikiCS** (node classification) and **COLLAB** (graph classification), varying the random walk length in [4, 8, 16, 32, 64]. As shown in the table below, while standard models like GraphSAGE and GIN suffer significant performance degradation as depth increases, GPM maintains strong performance, with only a slight drop at extreme walk lengths. We attribute this to GPM's architecture, where a **Transformer processes each sampled walk independently**, effectively decomposing hop-level interactions, rather than stacking layers that prioritize low-order information.
>
> | **Model Layer / Walk Length** | **4** | **8** | **16** | **32** | **64** |
> | --- | --- | --- | --- | --- | --- |
> | **WikiCS (Node classification)** |  |  |  |  |  |
> | SAGE | 78.50 | 73.38 | 68.35 | 64.32 | 30.58 |
> | GPM (ours) | 79.64 | **80.19** | 80.52 | 78.98 | 77.31 |
> | **COLLAB  (Graph Classification)** |  |  |  |  |  |
> | GIN | 80.31 | 79.30 | 75.44 | 72.68 | 65.38 |
> | GPM (ours) | 80.45 | **80.70** | 80.66 | 79.53 | 77.39 |
>
> > Experiments on heterogeneous graphs
> >
>
> We appreciate the suggestion and have conducted additional experiments on heterogeneous graphs. Specifically, we evaluated GPM on two widely used datasets: **ACM** and **DBLP**. To handle heterogeneity, we adopt a simple yet effective strategy: for each node type, we use a **type-specific Mapping** to project the features into a shared latent space, following the approach in [1].
>
> We compare GPM with strong baselines including Metapath2vec, RGCN, HAN, HeCo, and HGT. Using the standard 20/40/40 split and Micro-F1 as the metric, GPM outperforms all baselines:
>
> | **Dataset** | **Metapath2vec** | **HAN** | **HeCo** | **HGT** | **GPM (ours)** |
> | --- | --- | --- | --- | --- | --- |
> | ACM | 82.96 | 90.45 | 91.23 | 91.12 | **93.27** |
> | DBLP | 89.02 | 92.60 | 93.24 | 92.55 | **94.29** |
>
> [1] Are we really making much progress? Revisiting, benchmarking, and refining heterogeneous graph neural networks, KDD 21.
>
> > Experiment on large graphs, like OGB.
> >
>
> Thank you for pointing this out. In fact, we already include experiments on large-scale **OGB benchmarks** in the main paper: **ogbn-arxiv (Arxiv)**, **ogbn-products (Products)**, and **ogbl-collab**. The **ogbn-products** dataset, for instance, includes **2,449,029 nodes** and **123,718,024 edges**. Despite the scale, GPM maintains high scalability and **consistently outperforms baselines** across these large benchmarks. For example, GPM achieves 82.62 in ogbn-products yet the best baseline just achieves 82.00.
>
> > Strongly recommend the author publicly disclose the code to facilitate reproducibility.
> >
>
> We completely agree, and we are fully committed to open research. We have provided the code in the supplement. We will release the full source code and instructions upon publication.

---

> > ### Comment · Reviewer_Ynb7 · 2025-04-02
> >
> > Thanks for the authors' reply, I'd like to maintain my score.

---

> > > ### Author Response · Authors · 2025-04-06
> > >
> > > Thank you for your response. Your feedback is genuinely appreciated and helps us continue to improve our work.

---

### Official Review · Reviewer_M2he · 2025-03-10

**Overall Recommendation:** 2

**Summary:**

This paper proposes GPM, a graph transformer based on randomly sampling walks in the graph. GPM achieves strong empirical results across a variety of datasets and types of tasks. Furthermore, by sampling a large number of paths (or very long paths) GPM achieves high expressivity.

## update after rebuttal
The authors did provide substantial explanations and additional experiments during the rebuttal. This has caused me to increase my score (see latest rebuttal comment).

**Claims And Evidence:**

The experimental claims are solid, the evidence supporting theoretical claims could be improved (see below).

**Essential References Not Discussed:**

The expressivity of of GPM is based on randomly sampling graph patterns. It seems that (1) is essential related work, as (1) propose expressive GNNs based on randomly sampled patterns (homomorphisms). Furtherme, since the architecture is based on walks I believe that recent literature on path-based GNNs should also be discussed such as (2,3).

(1) Welke at al., _Expectation-Complete Graph Representations with Homomorphisms_, ICML 2023
(2) Gaspard et al., _Path Neural Networks: Expressive and Accurate Graph Neural Networks_, ICML 2023
(3) Drucks et al.,  _The Expressive Power of Path based Graph Neural Networks_, ICML 2024

**Experimental Designs Or Analyses:**

This paper very thoroughly evaluates their methods for different tasks and datasets. Overall, I think that the experiments are solid (see Strengths) except the question I raise in (Questions For Authors).

**Methods And Evaluation Criteria:**

See below.

**Other Comments Or Suggestions:**

- A graph is defined as $G = (V, E)$, this should also include the node features. Since the paper also mentions edge feature, they should also be defined here.
- Definition 3.1 seems incomplete as the $pos$ function is not defined

**Other Strengths And Weaknesses:**

**Strengths:**
- (S1). The (non mathematical) sections are well written and easy to follow.
- (S2). The proposed method is conceptually elegant and the paper has many small, clever ideas. In particular, the test time augmentation (only sampling few paths during training to speed it up but sampling a lot of paths to achieve strong inference results) seems really interesting.
- (S3). The experiments are more than solid. The authors select a wide range of different tasks (node predictions, link predictions, graph predictions, synthetic graph distinction) and achieve good results. All in all, I believe that the experimental part of this paper is excellent.

**Weaknesses:**
- (W1). Theoretical runtime. The proof of Proposition 3.2 requires sampling as many paths as the size of the $k$-hop neighborhood (for each node). Sampling this many paths for every node is computationally expensive (at least quadratic in the worst case).
- (W2). Mathematical precision. Definitions are incomplete (see below), Propositions and Theorems are formulated descriptively but are not precise mathematical statements.  Some examples of this:
	- Proposition 3.2 and 3.3 are informal.
	- Theorem 3.4: "GPM can distinguish non-isomorphic graphs given a sufficient number of graph patterns." This formulation is wrong: MPNNs are also able to distinguish non-isormophic graphs (just not all of them). I think what was meant is: "GPM can distinguish all pairs of non-isomorphic graphs given a sufficient number of patterns".
	- If my interpretation of Theorem 3.4 is correct, then Theorem 3.5 should be a simply corrollary of Theorem 3.4 instead of a theorem.
	- Theorem 3.4 and 3.5 rely on the reconstruction conjecture to be true which I believe is not mentioned in the main text.
	- Proof of Theorem 3.4: _"The simplified case assumes $n = 1$ with a sufficiently large walk length $l$, whereas GPM operates with $n > 1$. Since random walks can start from any node in the graph, an $l$-length random walk can be split into $k$ segments ($k$ is large enough), each of length $l/k$. Each sub-walk can be encoded individually and later combined to form the final embedding"_. I do not think long walks can simply be reconstructed from shorter walks as anonymous walks do not allow to match nodes between different walks.


**Overall,** this is a difficult decision for me. The experimental side of this paper is strong and I wish all GNN papers would so thoroughly evaluate their architecture on different tasks. However, the mathematical / theoretical side needs extensive improvements. Thus, I am in favor of rejecting as I believe that the required changes are too large to be done in the process of the rebuttal. I hope the authors will re-write the problematic sections and look forward to seeing this paper at a future conference.

**Questions For Authors:**

- For the "Empirical expressiveness analysis" (Tab 8 Appendix), did you ensure that the same patterns get sampled for pairs of graphs that you try to distinguish? If not, then your model would be trivially be able to distinguish 100% of all pairs.

**Relation To Broader Scientific Literature:**

I think the related work is mostly fine, but is missing a few references (see below).

**Theoretical Claims:**

I have skimmed all proofs in the appendix. While they seem to hold, mathematical statements are often imprecise (see below).

---

> ### Author Rebuttal · Authors · 2025-03-30
>
> We sincerely appreciate your recognition of the strengths of our method and experimental evaluation. While we take your concerns about the theoretical aspects seriously, we would like to emphasize that the primary contribution of our work lies in the **model design and empirical success**; the theoretical components are included to provide **insight into our design choices**, rather than to serve as the core contribution. We address each point in detail below and are committed to improving clarity and rigor in future revisions.
>
> > Related works on graph expressiveness
> >
>
> Thank you for pointing out these relevant works. While GPM shares high-level similarities with [1], it differs significantly in approach: GPM is a **data-driven framework** that learns task-relevant patterns through sampling and optimization, whereas [1] takes a **theoretical route**, leveraging homomorphism counts to prove expectation-completeness using random subgraph probes.
>
> Compared to PathNN [2] and PAIN [3], which also rely on random walks, GPM distinguishes itself by using a **Transformer to aggregate pattern-level embeddings**, rather than encoding enumerated paths [2] or sampled paths [3]. This enables **pattern-centric reasoning** and contributes to improved **generalization** across tasks and graph scales.
>
> Empirically, GPM outperforms all three baselines on the ZINC regression benchmark, evaluated via MAE:
>
> | **Dataset** | **GIN + hom + F [1]** | **PathNN-AP [2]** | **PAIN [3]** | **GPM (ours)** |
> | --- | --- | --- | --- | --- |
> | ZINC | 0.174 | 0.090 | 0.148 | **0.064** |
>
> [1] Expectation-Complete Graph Representations with Homomorphisms
>
> [2] Path Neural Networks: Expressive and Accurate Graph Neural Networks
>
> [3] The Expressive Power of Path based Graph Neural Networks
>
> > Theoretical runtime
> >
>
> We appreciate the reviewer raising this point. Indeed, in the worst-case scenario (e.g., dense graphs), the number of paths within a k-hop neighborhood can grow exponentially with k, making full enumeration intractable. However, **GPM does not require exhaustive enumeration**. Instead, we perform **efficient random walk sampling** to approximate the distribution over substructures:
>
> - In practice, we sample a fixed number of paths per node (e.g., 128) of fixed length (e.g., 8), independent of the total k-hop neighborhood size. This provides predictable runtime and memory cost, even on large-scale graphs. Notably, sampling random walks on graphs with over **2.5 million nodes** in under two minutes.
> - Theoretical analysis in Proposition 3.2 assumes access to the full pattern space, but empirically we find that a **small number of walks (i.e., 128)** is sufficient for strong performance.
>
> > Mathematical precision
> >
>
> We thank the reviewer for the detailed feedback. We agree that the current formulation of Propositions 3.2, 3.3, and Theorems 3.4 and 3.5 is informal and lacks full mathematical rigor. These results were intended to provide **conceptual insights** into the representational power of GPM, not to constitute formal theoretical contributions.
>
> That said, we will: (1) **Revise Theorem 3.4** to state clearly that *GPM can distinguish all pairs of non-isomorphic graphs given a sufficiently rich set of patterns*. (2) **Demote Theorem 3.5** to a corollary or informal observation, as it logically follows from Theorem 3.4. (3) Explicitly **mention the reconstruction conjecture** and its role in the assumptions underlying these results.
>
> Regarding the proof of Theorem 3.4: we agree that our phrasing was misleading. We did **not mean to suggest** that long walks can be **fully reconstructed** from shorter ones, especially since anonymous walks cannot preserve node identity across segments. Rather, we intended to describe an **approximate strategy**, where **long walks are segmented into shorter sub-walks**, each encoded independently. This design allows the transformer to aggregate distributed long-range information across these sub-patterns. We will revise the wording in the appendix to reflect this more accurately and avoid overstating the implications.
>
> To reiterate, our theoretical analysis is included to motivate and clarify design choices. We will clearly label these components as **heuristic or informal** in future revisions to avoid confusion.
>
> > Other modifications and clarifications
> >
>
> **Modifications**: We thank the reviewer for pointing this out. We will update the definition of a graph and the definition 3.1 in our future version.
>
> **Clarifications**: We appreciate the question about sampling consistency in the expressiveness analysis (Table 8). To clarify: GPM samples patterns **locally from each graph**, with the goal of capturing the **structural fingerprint** of each instance via its own distribution of substructures. **different patterns are not a source of trivial separability**, but rather a core mechanism for capturing graph-specific structure. Using a shared pattern set would **reduce discriminative power**.

---

> > ### Comment · Reviewer_M2he · 2025-04-04
> >
> > I thank the authors for the reply. I believe these changes will improve the paper and thus will slightly improve my score (reject -> weak reject).

---

> > > ### Author Response · Authors · 2025-04-06
> > >
> > > Thank you for your response and for raising the score. Your feedback is genuinely appreciated and helps us continue to improve our work.

---

### Official Review · Reviewer_uZYz · 2025-03-13

**Overall Recommendation:** 3

**Summary:**

This paper introduces a framework for graph learning named Neural Graph Pattern Machine (GPM). It aims to directly capture substructure patterns instead of relying on message-passing mechanisms. The model leverages random walks to extract graph patterns, and then convert them into semantic paths and anonymous paths. These paths are encoded separately through a Transformer-based architecture to learn deep representations. Empirical evaluations are conducted across multiple tasks, including node classification, link prediction, and graph regression. The experimental results can demonstrate GPM’s superiority over baseline models including message-passing and other Transformer-based models. The paper also highlights its robustness to distribution shifts, scalability, and interoperability.

## update after rebuttal
Thank you for the authors' rebuttal. I will keep my score.

**Claims And Evidence:**

Strengths:
1. The advantages of the proposed GPM are empirically supported by experimental results on node/edge/graph learning tasks. The method is computationally efficient compared to the naive GCN.
2. The paper is well-structured and clearly written. The authors effectively explain their motivation, methodology, and results.

**Essential References Not Discussed:**

See above [1,2].

**Experimental Designs Or Analyses:**

Strengths:
1. The experimental evaluation is thorough, covering multiple tasks including node classification, link prediction, and graph regression. A wide range of baseline models are compared.
2. The proposed GPM demonstrates strong empirical results across various datasets, outperforming or matching state-of-the-art models in benchmark tasks. These results suggest that the method is competitive and effective in real-world applications.

Weaknesses:
1. The missing numbers in Table 1 are not explicitly explained in the text.

**Methods And Evaluation Criteria:**

Strengths:
1. The method is both conceptually sound and practically feasible
2. The evaluation criteria follow common practices and make sense.

**Other Comments Or Suggestions:**

NA

**Other Strengths And Weaknesses:**

NA

**Questions For Authors:**

Please address the Weaknesses listed above.

**Relation To Broader Scientific Literature:**

Strengths:
1. A wide range of existing methods are discussed including both message-passing based and tokenization based models.

Weaknesses:
1. Extracting subgraph patterns with random walk and anonymous walk have been studied in previous papers [1, 2]. [1] was published on the recent ICLR’25 but was posted on arxiv on Jun, 2024. [2] studied anonymous walks for temporal networks which are based on GNNs. These papers should be discussed and compared as well since the main idea of GPM seems very relevant to them.
2. The proposed Important Pattern Identifier is a straightforward application of Transformer interpretability. This part doesn’t show a significant technical contribution.

[1] Learning Long Range Dependencies on Graphs via Random Walks. (ICLR’25)
[2] Inductive Representation Learning in Temporal Networks via Causal Anonymous Walks. (ICLR’21)

**Theoretical Claims:**

Strengths:
1. The paper provides a comprehensive theoretical study. These theoretical insights help establish the foundations of the proposed method and its advantages over traditional message-passing GNNs.

Weaknesses:
1. The statement of proposition 3.3 “provide a comprehensive representation…” lacks rigor. The term “comprehensive” is a subjective description. Also, I didn’t understand how the proof in Appendix C.2 proves this claim.

---

> ### Author Rebuttal · Authors · 2025-03-30
>
> We sincerely thank the reviewer for their detailed and insightful feedback, as well as for recognizing our contributions in methodology, theoretical insights, and empirical performance. We particularly appreciate the constructive suggestions regarding theoretical clarity, experimental completeness, and related work comparisons, which have helped us significantly improve the manuscript.
>
> > Discussion and comparison to related works.
> >
>
> We highlight several key differences between GPM and NeuralWalker: **(1) Task-adaptive tokenization**: NeuralWalker tokenizes graphs into random walks uniformly across all tasks. In contrast, GPM performs **task-specific pattern sampling**—for example, node-level tasks focus on patterns centered around each node, while graph-level tasks use global structures. This adaptivity is a central design of GPM. **(2)** **Representation granularity**: NeuralWalker encodes each node within a walk and updates node-level embeddings, resulting in multiple embeddings per walk. GPM, on the other hand, produces a **single representation per walk**, capturing the semantics of the entire pattern rather than individual nodes. **(3) Architecture paradigm**: GPM is a **fully message passing-free** architecture, whereas NeuralWalker still relies on message passing components to explicitly model local information.
>
> Regarding CAW, both CAW and GPM use anonymous paths as substructure patterns. However, their purposes and applications differ: **(1) Domain**: CAW is designed for **temporal graphs**, while GPM is intended for **attributed static graphs**. **(2) Tokenization**: Like NeuralWalker, CAW performs **uniform walk sampling**, regardless of the task. GPM introduces an **adaptive tokenizer** tailored to different downstream objectives. **(2) Task scope**: CAW primarily focuses on **link prediction**, while GPM supports **node, link, and graph-level tasks**.
>
> We summarize the empirical comparison between GPM and NeuralWalker in the following table. Since CAW is designed for temporal graphs, it does not naturally apply to the datasets used in this paper:
>
> | **Task** | **Dataset** | **NeuralWalker [1]** | **CAW [2]** | **GPM (ours)** |
> | --- | --- | --- | --- | --- |
> | Node Classification | WikiCS | 78.55 | - | **80.19** |
> | Link Prediction | Cora | 87.50 | - | **92.85** |
> | Graph Regression | ZINC | **0.053** | - | 0.064 |
>
> [1] Learning Long Range Dependencies on Graphs via Random Walks. (ICLR’25)
>
> [2] Inductive Representation Learning in Temporal Networks via Causal Anonymous Walks. (ICLR’21)
>
> > Elaborate on Proposition 3.3.
> >
>
> We apologize for the confusion and appreciate the opportunity to clarify. In Proposition 3.3, the term *"comprehensive"* refers to the ability to capture both **semantic (feature-level)** and **structural (topological)** information of a graph pattern by jointly modeling its **semantic path** and **anonymous path**. Specifically, a graph pattern can be decomposed into a semantic path (preserving node/edge attributes) and an anonymous path (preserving topology only). These paths, being sequential, can be bijectively projected into fixed-size embeddings without loss of information. Therefore, embedding a graph pattern can be obtained by combining the embeddings of its semantic and anonymous paths.
>
> This proposition is intended to provide the intuition behind GPM’s dual encoding design and emphasizes the necessity of modeling both views to capture rich structural and semantic signals.
>
> > The Important Pattern Identifier is a straightforward application of Transformer interpretability.
> >
>
> We appreciate this feedback and would like to clarify the intent of this module.
>
> This component serves primarily as an **aggregation mechanism** over encoded graph patterns. We chose to use the Transformer for this role due to its proven effectiveness across domains, e.g., CV and NLP. While we acknowledge that Transformer attention is not novel per se, its **emergent interpretability**—i.e., its ability to highlight important substructures—was a valuable and interpretable byproduct. This insight supports GPM’s modular design, where effective aggregation and interpretability naturally align.
>
> > The missing numbers are not explicitly explained in the text.
> >
>
> Thank you for pointing this out. We will update the text to clarify that missing values (denoted as “-”) correspond to baselines for which: **(1)** **Computational constraints** (e.g., tuning models such as VCR-Graphormer, GEANet) made training impractical, or **(2) Reproducibility issues** (e.g., unavailable official code for RAW-GNN, GraphMamba, GCFormer) prevented fair comparison.
>
> We prioritized reporting results for the most informative and representative baselines under a consistent evaluation protocol.

---

### Decision · Program_Chairs · 2025-05-01

**Decision:**

Accept (poster)

**Comment:**

This paper investigates the problem of identifying fundamental substructures within graphs using graph learning models. To address this challenge, the authors propose Neural Graph Pattern Machine (GPM), a framework designed to learn directly from graph patterns. GPM efficiently extracts and encodes substructures, while identifying the most relevant ones for downstream tasks. Experiments on several datasets demonstrate the effectiveness of the proposed model.


Strengths:
1. The paper is well-structured and clearly written. The authors effectively explain the motivation, methodology, and results.
2. The proposed method is both conceptually sound and practically feasible.
3. The experiments are sufficiently comprehensive to demonstrate the effectiveness of the proposed model.


Weaknesses:
1. The theoretical runtime analysis may suggest a high computational cost, and the mathematical precision could be improved in some areas.
2. Several related studies should be discussed and potentially compared in the paper to provide a more complete context for the proposed work.



Overall, this paper presents a novel and interesting approach to identifying fundamental substructures in graphs, making valuable contributions to the research field. The paper is well-written, clear, and easy to follow, and the experiments are sufficient to demonstrate the effectiveness of the proposed model. However, the paper would benefit from a more precise explanation of certain mathematical statements to enhance clarity. Additionally, some related studies should be discussed and even compared to strengthen the paper's argument. I recommend that the authors address these issues in the camera-ready version to make the paper more convincing.